# A mammalian-like piRNA pathway in Axolotl reveals the origins of piRNA-directed DNA methylation

Xinyu Xiang[1,2,3], Anni Gao [ID][3], Dominik Handler [ID][4], Francisco Falcon[5,6], Diego Rodriguez-Terrones[5], Sergej Nowoshilow [ID][5], Wanlu Liu [ID][3,7,8], Elly M Tanaka [ID][5] & Dónal O'Carroll [ID][1,2][✉]

## Abstract

The piRNA pathway protects animal germlines from active transposons. Mammals employ a cytoplasmic pathway to destroy transposon transcripts during germline reprogramming. This post-transcriptional mechanism is ancient and found throughout the animal kingdom. A nuclear piRNA pathway mediates transposon DNA re-methylation, which is believed to be bespoke to mammals. However, when exactly piRNA-directed DNA methylation evolved remains unknown. We found that a mammalian-like piRNA pathway evolved early in tetrapod evolution and is found and expressed in its current configuration in the axolotl salamander. Analysis of axolotl testes and oocytes revealed diverse repertoires of piRNAs and pervasive post-transcriptional targeting of young transposons. We identified high levels of genome methylation in axolotl spermatozoa, with full-length transposons being heavily methylated. Our findings reveal that the mammalian nuclear piRNA pathway has ancient vertebrate origins, and it has likely been safeguarding the germline throughout most of tetrapod evolution. Thus, the emergence of piRNA-directed DNA methylation is a pivotal epigenetic evolutionary event that may have laid the foundation for germline reprogramming and genomic imprinting.

**Keywords** piRNA; Germline; Transposon; DNA Methylation; Germline Reprogramming; Genomic Imprinting
**Subject Categories** Evolution & Ecology; RNA Biology

## Introduction

The germline is the cell lineage that gives rise to the gametes and ensures the continuity of life. Germline specification during embryonic development occurs by two distinct mechanisms: epigenesis and preformation. Epigenesis is the ancestral mode of germ cell acquisition, in which primordial germ cells (PGCs) are extrinsically induced from pluripotent cells by signals from surrounding tissues (Extavour and Akam, 2003). In contrast, preformation, which has independently evolved multiple times during evolution, involves the autonomous specification of PGCs by maternally derived molecules known as germ plasm (Mahowald, 1977). The fundamental distinction between these two mechanisms is that preformation allows direct germline continuity through germ plasm and occurs early in embryogenesis, while epigenesis relies on inductive signals to specify PGCs later during embryogenesis (Lawson et al, 1999). The convergent evolution of preformation suggests potential selective advantages. It has been hypothesized to confer increased evolvability (Johnson and Alberio, 2015), the capacity to generate heritable, selective, and phenotypic variation (Kirschner and Gerhart, 1998). Among tetrapods, germline specification predominantly follows epigenesis, with the exception of frogs and birds, which utilize preformation (Extavour and Akam, 2003).

Active transposons pose an existential threat to the integrity and continuity of the germline. The mammalian germline is particularly vulnerable during the period of germline reprogramming when transposon-repressing DNA methylation is erased (Seisenberger et al, 2012; Greenberg and Bourc'his, 2019) and transposon expression is unleashed (Bourc'his and Bestor, 2004; Aravin et al, 2007; Carmell et al, 2007). When expressed, active transposons can integrate into new genomic locations, potentially causing gene mutations (Bourque et al, 2018). In addition, transposition is also a major source of DNA damage (Gasior et al, 2006; Farkash and Prak, 2006; Belgnaoui et al, 2006). Given that germ cells are exquisitely sensitive to DNA damage, transposon-inflicted DNA double-strand breaks often result in germ cell death (Goodier and Kazazian, 2008). Indeed, failure to repress transposon activity in the germline broadly results in defective gametogenesis and infertility (Carmell et al, 2007; Bourc'his and Bestor, 2004). That said, transposition does occur in the germline at a very low frequency. For instance, in humans, approximately one in every 200 live births carries a novel LINE1 transposition event (Xing et al, 2009). These insertions are often non-deleterious and can generate

[1]Centre for Regenerative Medicine, Institute for Regeneration and Repair, Institute for Stem Cell Research, University of Edinburgh, 5 Little France Drive, Edinburgh EH16 4UU, UK. [2]Centre for Cell Biology, University of Edinburgh, Michael Swann Building, Max Born Crescent, Edinburgh EH9 3BF, UK. [3]Zhejiang University-University of Edinburgh Institute (ZJU-UoE Institute), Zhejiang University School of Medicine, International Campus, Zhejiang University, 314400 Haining, China. [4]Institute of Molecular Biotechnology of the Austrian Academy of Sciences (IMBA), Vienna BioCenter (VBC), Campus Vienna Biocenter, Vienna 1030, Austria. [5]Institute of Molecular Pathology (IMP), Vienna Biocenter (VBC), Campus Vienna Biocenter, Vienna 1030, Austria. [6]Vienna BioCenter PhD Program, Doctoral School of the University of Vienna and Medical University of Vienna, Vienna, Austria. [7]Department of Rheumatology and Immunology of the Second Affiliated Hospital, Zhejiang University School of Medicine, Zhejiang University, 310009 Hangzhou, China. [8]Dr. Li Dak Sum & Yip Yio Chin Center for Stem Cell and Regenerative Medicine, Zhejiang University, 310058 Hangzhou, China. ✉E-mail: donal.ocarroll@ed.ac.uk

selectable genetic variation, providing a potential source for genomic innovation. However, genomes expand when transposons excessively proliferate. An extreme example of genome expansion in vertebrates is the lungfish, a lobe-finned fish with a genome size ranging from 30 to 90 Gb and 60–90% of transposon-derived content (Wang et al, 2021; Schartl et al, 2024). Similarly, the axolotl salamander (*Ambystoma mexicanum*) possesses a huge 32 Gb genome with ~70% transposon content (Nowoshilow et al, 2018). Interestingly, such genomic gigantism and high transposon burden in vertebrates appear to occur in these water-to-land transition species (Falcon et al, 2023).

While a significant portion of vertebrate genomes originates from transposons, only a tiny fraction of transposon loci retain transposition activity, with the vast majority existing as dead mutated copies (Bourque et al, 2018). For example, only ~100 copies of LINE1 remain active in humans (Brouha et al, 2003), although 53% of the genome is derived from transposons. Time and transposon silencing ultimately kill active transposon copies and eventually transposon families. Over time, transposons accumulate inactivating mutations, and as such they can only survive if they manage to transpose in the germline. So, the fight for transposon survival and the opposing genomic battle to tame transposons occurs in the germline. The PIWI-interacting RNA (piRNA) pathway protects metazoan genomes and germlines from active transposons (Ozata et al, 2019; Wang et al, 2023). Not only does it acutely protect the germline, but it is also the principal transposon-taming force.

At the core of the piRNA pathway are the piRNAs and the PIWI proteins (Ozata et al, 2019; Wang et al, 2023). piRNAs are short non-coding RNAs that are bound to PIWI proteins. Through base complementarity, piRNAs act as guides to recruit the PIWI proteins to cellular RNAs. The piRNA pathway consists of two major components: a cytoplasmic and a nuclear pathway. The cytoplasmic piRNA pathway is responsible for piRNA biogenesis and post-transcriptional transposon silencing (Ozata et al, 2019; Wang et al, 2023). In vertebrates, the cytoplasmic PIWI proteins, PIWIL1 and PIWIL2, are piRNA-guided endonucleases that cleave transposon transcripts to suppress transposition (Houwing et al, 2008; De Fazio et al, 2011; Reuter et al, 2011; Houwing et al, 2007). On the other hand, the nuclear piRNA pathway mediates transcriptional silencing of transposons through DNA methylation and chromatin modifications (Ozata et al, 2019; Wang et al, 2023). The cytoplasmic piRNA pathway is common to most animals (Grimson et al, 2008). In vertebrates, the exoribonuclease PNLDC1 (PARN Like Ribonuclease Domain Containing Exonuclease 1) is responsible for trimming the 3′ of piRNA precursors to their mature length during piRNA biogenesis (Zhang et al, 2017; Ding et al, 2017; Nagirnaja et al, 2021). However, invertebrates apply different enzymes to trim piRNAs, such as Trimmer in silkworms (Izumi et al, 2016), PARN-1 in *C. elegans* (Tang et al, 2016), and Nibbler in flies (Hayashi et al, 2016; Han et al, 2011; Liu et al, 2011). Although these enzymes share the common function of processing piRNA 3′ termini, their evolutionary divergence reflects lineage-specific adaptations in piRNA biogenesis between vertebrates and invertebrates. The nuclear piRNA pathway has also arisen independently during evolution (Ozata et al, 2019; Wang et al, 2023) and is not present in all animals. Worms, flies, and mammals utilize piRNA-guided nuclear PIWI proteins to repress transposon transcription; however, the downstream factors and molecular mechanisms are distinct (Ozata et al, 2019; Wang et al, 2023). In *C. elegans* and *Drosophila*, transposon silencing is achieved through histone methylation and heterochromatinization (Rangan et al, 2011; Wang and Elgin, 2011; Klenov et al, 2011; Sienski et al, 2012; Shirayama et al, 2012; Ashe et al, 2012; Luteijn et al, 2012). In contrast, the mammalian nuclear piRNA pathway primarily silences young active transposons through DNA methylation during the process of de novo genome methylation that follows germline reprogramming (Aravin et al, 2007; Carmell et al, 2007; Kuramochi-Miyagawa et al, 2008). piRNA-directed LINE1 methylation is installed by a two-step authentication process (Dias Mirandela et al, 2024). Firstly, chromatin modification recruits the piRNA factor SPOCD1 through the chromatin reader SPIN1 to transposon promoters (Dias Mirandela et al, 2024). Secondly, through base complementarity piRNAs tether PIWIL4 (MIWI2) to nascent transposon transcripts (De Fazio et al, 2011; Schöpp et al, 2020). The outcome of these events is the recruitment of the de novo DNA methylation machinery (DNMT3L and DNMT3B/C) and piRNA-guided transposon methylation (Bourc'his and Bestor, 2004; Barau et al, 2016; Jain et al, 2017; Zoch et al, 2020, 2024; Stallmeyer et al, 2024). This silencing mechanism also involves other factors such as C19ORF84 (Zoch et al, 2020, 2024) and TEX15 (Yang et al, 2020; Schöpp et al, 2020). In summary, the factors PIWIL4, SPOCD1, SPIN1, DNMT3B/C, C19ORF84, and TEX15 that mediate piRNA-directed DNA methylation define the mammalian nuclear piRNA pathway. Currently, the piRNA-mediated transposon methylation mechanism has only been reported in mammals, with its true origin remaining unknown.

## Results

### Axolotl has a mammalian-like piRNA pathway

To investigate the origin of the modern mammalian piRNA pathway, we searched for orthologues of mammalian piRNA factors throughout vertebrate evolution (Fig. 1A,B). An intact cytoplasmic piRNA pathway is present in all tetrapods and lobe-finned fish irrespective of germline specification (Fig. 1B; Dataset EV1). Ray-finned fish (Actinopterygii) genomes contain the *Piwil1* and *Piwil2* genes that encode the key cytoplasmic PIWI proteins responsible for piRNA-guided post-transcriptional transposon silencing and piRNA amplification (De Fazio et al, 2011; Reuter et al, 2011; Di Giacomo et al, 2013) (Fig. 1B). However, not all piRNA biogenesis factors are detected in ray-finned species. For example, both *Pnldc1* and *Mael* are absent in Actinopterygii genomes (Fig. 1B). The elephant shark representative of the cartilaginous fish contained all genes encoding the cytoplasmic piRNA pathway apart from *Pnldc1*. *Pnldc1* is first observed in Coelacanth as well as two lungfish species (Fig. 1B), and as such the modern form of the mammalian cytoplasmic piRNA pathway first evolved in the common ancestor of lobe-finned fishes.

We next analyzed the evolution of the nuclear branch of the mammalian piRNA pathway. *Piwil4*, which encodes the key nuclear PIWI protein, appears to have originated in the common ancestor of jawed vertebrates and is retained in cartilaginous fish such as the elephant shark, but has been lost in ray-finned fish (Gutierrez et al, 2021). Notably, *Piwil4* is present in lobe-finned fish and further

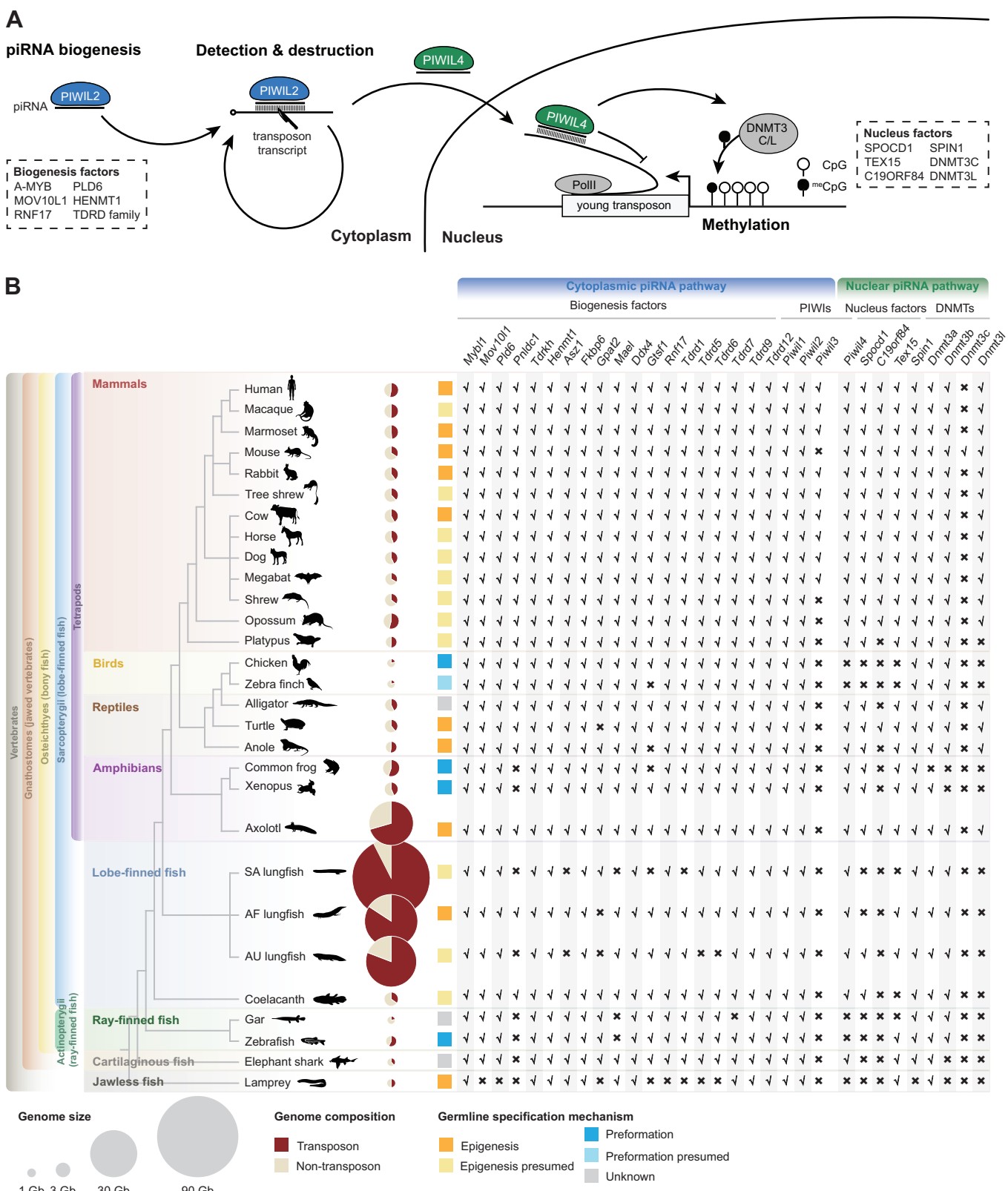

◄ **Figure 1. Modern mammalian piRNA pathway might originate from common tetrapod ancestors with the axolotl.**

(A) Schematic of piRNA pathway in mammals. Major piRNA biogenesis factors and nucleus factors are listed in box with dashed lines. (B) The piRNA pathway and germline specification mechanism for vertebrates. Left, phylogenetic tree of representative vertebrate species. Middle, genome size, transposon percentage, and the types of germline specification mechanism. Germline specification mechanisms for each species are from Extavour and Akam, 2003 and Hansen and Pelegri, 2021 (see "Methods"). Right, the existence of piRNA factors in vertebrate genomes. For lungfishes, South American lungfish (SA), African lungfish (AF), Australian lungfish (AU). Source data are available online for this figure.

retained across all tetrapods, except for birds (Fig. 1B). We next analyzed the recently identified PIWIL4-associated nuclear piRNA factors SPOCD1 (Zoch et al, 2020), C19ORF84 (Zoch et al, 2024), TEX15 (Schöpp et al, 2020; Yang et al, 2020) and SPIN1 (Dias Mirandela et al, 2024) (Fig. 1A). *Spocd1* is present in lobe-finned fish, including coelacanth and one lungfish genome (Fig. 1B). *Spocd1* is retained in all tetrapod genomes apart from birds. *C19orf84* is first observed in axolotl and also found in turtle and all mammals except for platypus (Fig. 1B). The *Spin1* gene is present in all jawed vertebrates. TEX15 is found in jawless and cartilaginous fish but lost in some ray- and lobe-finned fish (Fig. 1B), indicating an early origin in vertebrate evolution (Schöpp et al, 2023). Again, TEX15 is found in all tetrapods apart from birds. Next, we analyzed the de novo DNA methylation machinery. As previously reported (Barau et al, 2016; Jain et al, 2017), the *Dnmt3c* methyltransferase gene that arose from a gene duplication of *Dnmt3b* is muroid-specific (Fig. 1B). The de novo methyltransferase *Dnmt3* gene is ancient and arose in eukaryotic ancestors (Ponger and Li, 2005). *Dnmt3a* and *Dnmt3b* arose in the whole-genome duplication event in early vertebrate ancestors (Liu et al, 2020). We found the *Dnmt3a* gene in nearly all species analyzed except for frog (Fig. 1B), while *Dnmt3b* is not found in jawless or cartilaginous fish but is present in most other vertebrates except for frogs (Fig. 1B). The *Dnmt3l* gene encoding the adapter protein is not found in jawless, cartilaginous, ray- or lobe-finned fish. *Dnmt3l* is present throughout tetrapods but has been lost in birds, frogs, and platypus (Fig. 1B). In summary, the components of the nuclear piRNA pathway arose at different times in evolution, but they are found all together for the first time in axolotl, and strikingly they have been lost in birds.

## Axolotl piRNA pathways genes are predominantly expressed in the male and female axolotl germlines

Given that a mammalian-like piRNA pathway is first seen in axolotl, we next sought to explore the axolotl pathway and to search for evidence of its functionality. In mammals, most piRNA pathway factors' expression is restricted to the germline. The cytoplasmic piRNA pathway is expressed in both male and female human (Girard et al, 2006; Gainetdinov et al, 2017; Williams et al, 2015; Roovers et al, 2015; Yang et al, 2019), mouse (Kuramochi-Miyagawa et al, 2004; Deng and Lin, 2002; Flemr et al, 2013) and golden hamster (Loubalova et al, 2021; Ishino et al, 2021; Hasuwa et al, 2021; Zhang et al, 2021) germlines. In contrast, the nuclear piRNA pathway is restricted to the mammalian male germline (Carmell et al, 2007; Dias Mirandela et al, 2024; Aravin et al, 2008; Kuramochi-Miyagawa et al, 2008). We examined the expression of piRNA factors in axolotl tissues from previously published datasets (Bryant et al, 2017; Ye et al, 2022). Strikingly, the expression of the vast majority of piRNA factors is restricted to gonads (Fig. 2).

*Tdrd7* was an exception and broadly expressed in the soma (Fig. 2). *Spin1* was also expressed in the soma as is the case in mammals (The Human Protein Atlas, 2025; Mouse Genome Informatics Web Site et al, 2025) (Fig. 2). Both the cytoplasmic and nuclear piRNA pathways were expressed in both the male and female axolotl germlines. In summary, the expression of the axolotl piRNA pathway genes is mostly restricted to the germline.

## The landscape of active axolotl transposons

It is the active transposons that retain the ability to transpose which pose an acute threat to the germline and are the targets of the piRNA pathway. We therefore sought to identify which transposon families in the axolotl genome retain active copies. We applied the standard transposon annotation pipeline by RepeatMasker (Smit et al, 2013), with the unknown transposons identified by RepeatMasker further annotated by DeepTE (Yan et al, 2020) (see "Axolotl transposon annotation"). Approximately 70% of the axolotl genome is derived from transposon remnants (Nowoshilow et al, 2018). Among these, Long Terminal Repeats (LTRs) were the most abundant class of transposons (Nowoshilow et al, 2018) (Fig. 3A,B), comprising 24.5% of the genome (Fig. 3A,B). To roughly estimate which transposon copies could retain activity, we used the following criteria. Firstly, we selected copies with a length distribution close to the full-length consensus sequence (Dataset EV2). Copies with truncations have lost key elements or factors required for autonomous transposition. Secondly, we selected for young copies based on divergence from the consensus sequence. The accumulation of mutations over time often renders transposons inactive (Ozata et al, 2019). Transposon loci that were both full-length and young were defined as active copies. Using these criteria, for families where we could identify active copies, the frequency of active transposons within that family ranged between 0.003-0.6% (Figs. 3C and EV1A). This frequency translates into two active copies for LINE-I family to >13,000 for LTR-Gypsy family (Dataset EV2). Furthermore, for transposon families that have already proliferated in the axolotl genome, they had numerous active copies. To be specific, six families had more than 1000 likely active copies, encompassing LTR-Gypsy, LTR-ERV1, DNA-hAT, DNA-PIF-Harbinger, DNA-MULE, and DIRS (Fig. 3D; Dataset EV2). These six families constituted the majority of active transposons, accounting for 93.2% of all active copies (Fig. EV1B). Notably, the LTR class, which occupied the largest portion of the genome, had the highest proportion of likely active copies (0.21%, a total of 21,945 likely active copies), suggesting that LTRs might still be expanding in the axolotl genome (Fig. EV1C). In summary, we estimate that the vast majority of the full-length, close to consensus matching and thus likely active axolotl transposons are principally derived from six families and constitute less than 1% of total copies within those respective families.

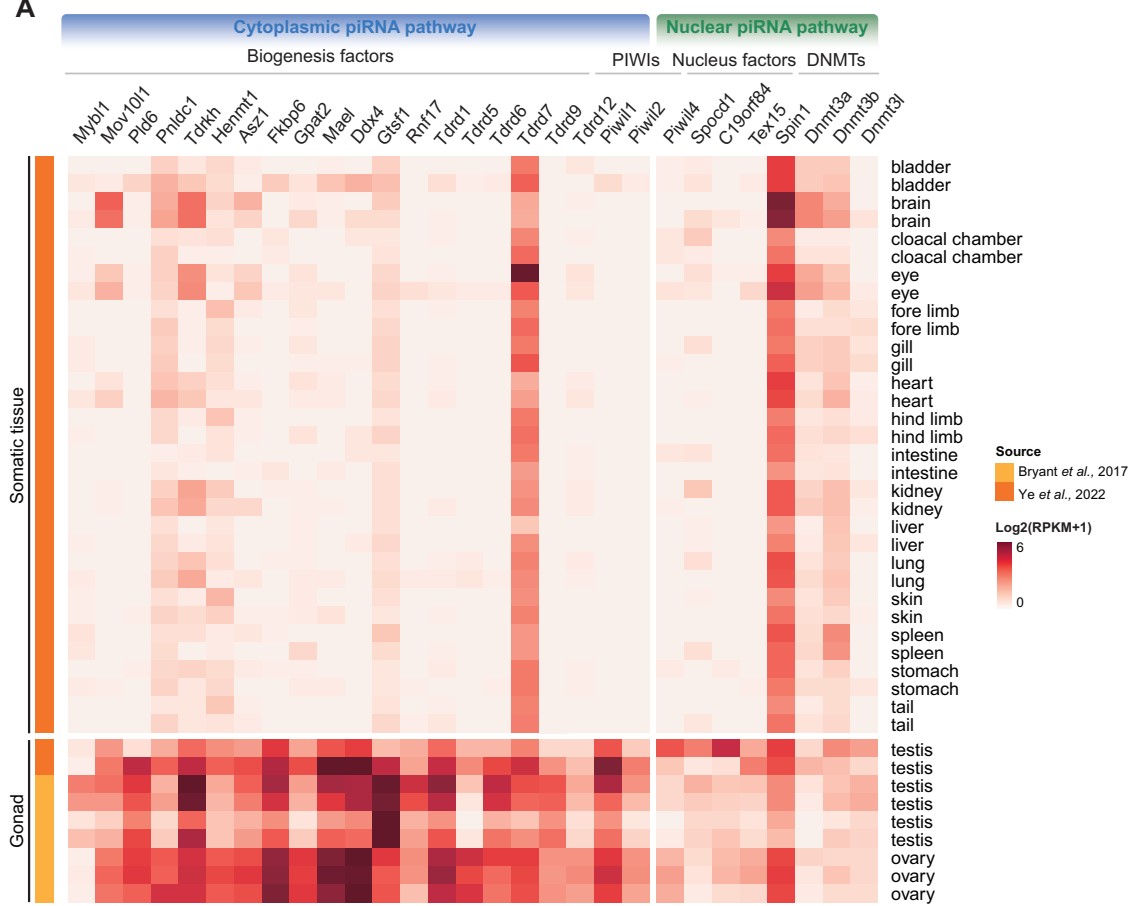

**Figure 2. The axolotl has a mammalian-like piRNA pathway.**

(A) Heatmap for expression of piRNA pathway factors in axolotl somatic and gonadal tissues. Source data are available online for this figure.

## The axolotl genome encodes a diverse repertoire of transposon-targeting piRNAs

Given the piRNA pathway is expressed in both the male and female axolotl germlines, using small RNA sequencing (sRNA-seq), we characterized axolotl piRNAs from two testis and oocyte samples (Fig. 4A; Dataset EV3). To focus on piRNAs, we examined small RNAs with a length greater than 23 nucleotides to exclude 19–22 nucleotide microRNAs (miRNAs) and short interfering RNAs (siRNAs) (Aravin et al, 2006; Girard et al, 2006; Vagin et al, 2006; Lau et al, 2006; Grivna et al, 2006; Saito et al, 2006). Our data captured the widespread existence of piRNAs, with a median length of 29 nt (Fig. 4A; Dataset EV3), consistent with recent findings (Schartl et al, 2024). piRNAs could be identified from both male and female axolotl germlines with the same length distribution, consistent with the piRNA pathway expression in both sexes (Fig. 2). A total number of 12–38 million distinct piRNAs could be identified among each of the samples (Fig. EV2A). Around 88 million unique piRNAs were identified across all samples in total (Fig. EV2A). Notably, piRNA sequences from different samples varied greatly, suggesting that the sequencing was not saturated (Fig. EV2A). We then mapped piRNAs to the genome and found that around 60–70% piRNAs were transposon-derived (Fig. 4B).

The annotation of piRNAs between male and female germline differed slightly (Fig. 4B). When mapping piRNAs to transposon families, all the active transposon families identified generate great quantities of piRNAs (Fig. 4C, Fig. EV2B). For example, the active transposon families LTR-ERV1, DNA-PIF-Harbinger and LINE-RTE were among the top five piRNA-producing families (Figs. 4C and EV2B). The piRNA signal over the consensus sequence of representative transposon families indicated that in both male and female germlines, piRNA mapped to sense and antisense strands across the entire consensus sequence (Figs. 4D and EV2C). The pattern of piRNA formation also differed between the sexes (Figs. 4D and EV2C).

We next identified piRNA clusters in the axolotl genome (Fig. 4E). Approximately 2500 piRNA clusters were identified in the male and female germlines (Fig. 4E). Ranking the piRNA clusters by their cumulative fraction of piRNA reads revealed axolotl piRNAs were predominantly derived from a small number of highly productive clusters (Fig. 4F), similar to observations in other vertebrate and invertebrate species (Konstantinidou et al, 2024). The top 500–1000 clusters accounted for 90% of all cluster-derived piRNAs (Fig. 4F). The majority of axolotl piRNA clusters were uni-strand (Fig. 4E), consistent with the unidirectional transcription mechanism typical of mammalian piRNA clusters like mouse

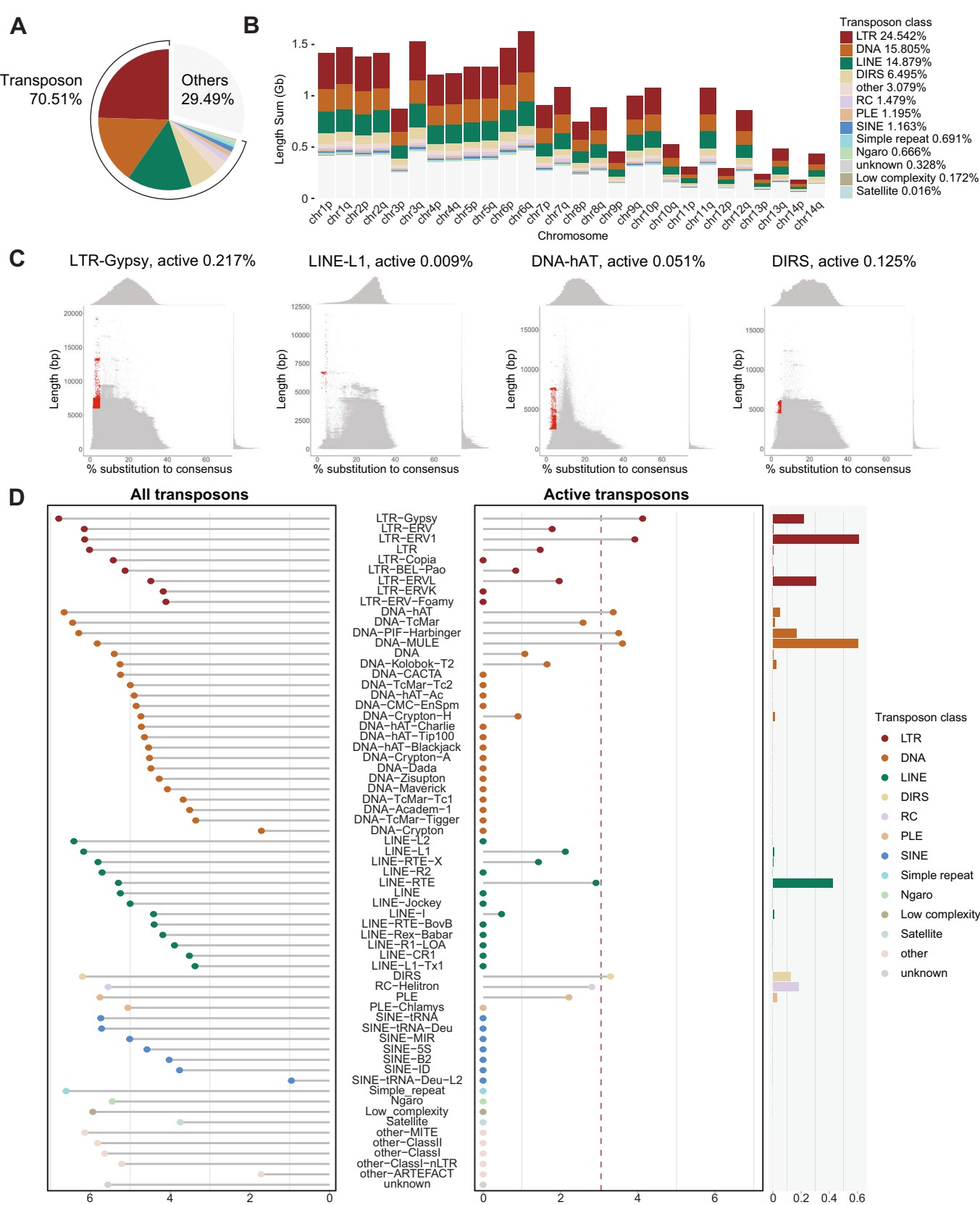

**Figure 3.   The landscape of potentially active axolotl transposons.**

(A, B) Genomic composition of transposon classes in axolotl. Both the whole genome (A) and each chromosome (B) are displayed. (C) Distribution of substitution rate and sequence length for every single copy in the representative family. Defined potentially active copies are highlighted in red. (D) Lollipop plots display the copy number for each transposon family. Total copy number (left) and potentially active copy number (right) are shown. The red dashed line indicates transposon families with copy numbers greater than 1000. Bar plot indicates potentially active copy number percentage within transposon family. Source data are available online for this figure.

(Girard et al, 2006; Aravin et al, 2006; Grivna et al, 2006; Li et al, 2013). Moreover, piRNA clusters in axolotl had a median length of 10 kb (Fig. 4G), a size distribution similar to mammals (Lau et al, 2006; Aravin et al, 2006; Girard et al, 2006; Grivna et al, 2006; Li et al, 2013). Collectively, the annotated piRNA clusters covered approximately 0.13% of the axolotl genome, with an average of 47% of piRNA sequences assigned to these clusters (Dataset EV4). Interestingly, while there is overlap between clusters expressed in the respective germlines, most clusters show sex-specific expression (Figs. 4H and 5A,B). Collectively, these data show abundant and diverse piRNA expression in both male and female axolotl germlines.

## The cytoplasmic axolotl piRNA pathway post-transcriptionally silences transposons

The abundance of piRNAs derived from active transposon families indicates a role for these piRNAs in post-transcriptional silencing through piRNA-mediated transposon transcript cleavage. The ping-pong cycle (Brennecke et al, 2007; Gunawardane et al, 2007; Aravin et al, 2007; De Fazio et al, 2011) or piRNA amplification (Fig. 6A) initiates when a complementary piRNA guides PIWI-mediated cleavage of the transposon transcript. Endonucleolytic PIWI-mediated cleavage is at the nucleotide opposed to the tenth nucleotide from the 5′ end of the piRNA. Primary piRNAs have a 5′U (1U) bias, therefore when a primary piRNA directs cleavage of a transcript followed by processing of the 3′ fragment into a secondary piRNA, this secondary piRNA will contain a bias for A at position 10 (10A) (Brennecke et al, 2007). Furthermore, there is also an overlap of 10 nucleotides between the primary and secondary piRNA pair (Brennecke et al, 2007). A second round of piRNA biogenesis mediated by secondary piRNAs results in the amplification of the initiating primary piRNAs. Thus, this amplification cycle not only destroys the transposon transcript to neutralize the acute threat of transposition but also serves to reinforce post-transcriptional silencing through the amplification of transposon-targeting piRNAs. The 10-nucleotide complementarity between amplification pairs as well as the 1U/10A bias can be used to determine the presence of piRNA amplification (Fig. 6A). A robust 10 nucleotide overlap between complementary piRNAs was observed across all classes of transposons in both the male and female germlines (Figs. 6B,D and EV3A). Furthermore, the bias for 10A and 1U was detected in both male and female transposon-related piRNA populations (Figs. 6C,E and EV3B). In summary, robust piRNA amplification is detected and the pathway post-transcriptionally silences transposons in both the adult male and female axolotl germlines.

## Evidence for an active nuclear piRNA pathway in axolotl

In mammals, piRNA-directed DNA methylation epigenetically silences young transposons (Aravin et al, 2007; Carmell et al, 2007;

Kuramochi-Miyagawa et al, 2008; Schöpp et al, 2020; Zoch et al, 2020, 2024; Dias Mirandela et al, 2024). Low-coverage maps of axolotl DNA methylation from somatic tissues exist (Klughammer et al, 2023), but they lack the resolution to confidently define features of genomic DNA methylation. To understand the characteristics of genome and transposon methylation in the male axolotl germline, we isolated spermatozoa and performed Enzymatic Methyl-seq (EM-seq) using biological triplicates. Each replicate had a depth/coverage of 1.5×/50% with a combined depth/coverage of 4.5×/77% (Dataset EV5), permitting a confident analysis of genome methylation patterns. In general, axolotl exhibited a relatively high CpG methylation level in sperm (~90%) (Fig. 7A), compared to human and mouse (75% and 85%, respectively) (Molaro et al, 2011; Hammoud et al, 2014). Methylation over axolotl gene bodies averaged around 80%, with a drop to 60% in transcription start site (TSS) regions (Fig. 7A,B). In comparison, the whole region of axolotl transposons was heavily methylated (Figs. 7C,D and EV4A). Active transposons showed higher methylation when compared to all copies, except for Rolling Circle (RC) transposons (Fig. 7C,D). Notably, the promoter regions of axolotl transposons were also heavily methylated, such as the 5′ flanking LTR repeats for LTR elements as well as the 5′ UTR for LINE elements (Fig. 7D).

To better understand the methylation landscape in axolotl, we also annotated CpG islands (CGIs) in axolotl. With standard filtering criteria by Gardiner and Frommer (Gardiner-Garden and Frommer, 1987), we identified 248,988 CGIs in axolotl, which mostly ranged from 20 to 1000 bp in length (Fig. EV4B). In mammals, CGIs are associated with low methylation levels (Illingworth and Bird, 2009; Deaton and Bird, 2011). In stark contrast, axolotl CGIs are associated with high methylation levels (Fig. 7A). Approximately 95% of CGIs are highly methylated in axolotl sperm, significantly higher than the genome level (Fig. 7E). Annotation revealed that the majority (93%) of CGIs in axolotl significantly overlapped with transposons rather than TSS/genic regions (Fig. 7F); which is the case for mammalian CGIs (Deaton and Bird, 2011; Smallwood et al, 2011; Sharif et al, 2010). Consistently, there was only a slight CGI enrichment within 100 bp TSS adjacent regions (Fig. EV4C). Notably, the methylation level of CGI-overlapped transposons was significantly higher than that of all transposons, but not significant at gene TSSs (Fig. 7G). This suggests that CGIs in axolotl play a role in transposon silencing. To be specific, all major transposon classes overlapping with CGIs exhibited higher methylation levels and less divergence from the consensus sequence (Figs. 7H and EV4D), indicating a role for CGIs in young transposon silencing. For DNA transposons and DIRS, ~60–80% of likely active copies were covered by CGIs (Figs. 7I and EV4E), implying potential regulation of DNA/DIRS activity. Taken together, we describe a confident map of axolotl genome methylation. The axolotl spermatozoan genome is highly methylated, and CGIs in axolotl are associated predominantly with transposons. In summary, promoter DNA methylation of active transposons is a feature of axolotl spermatozoan genome methylation.

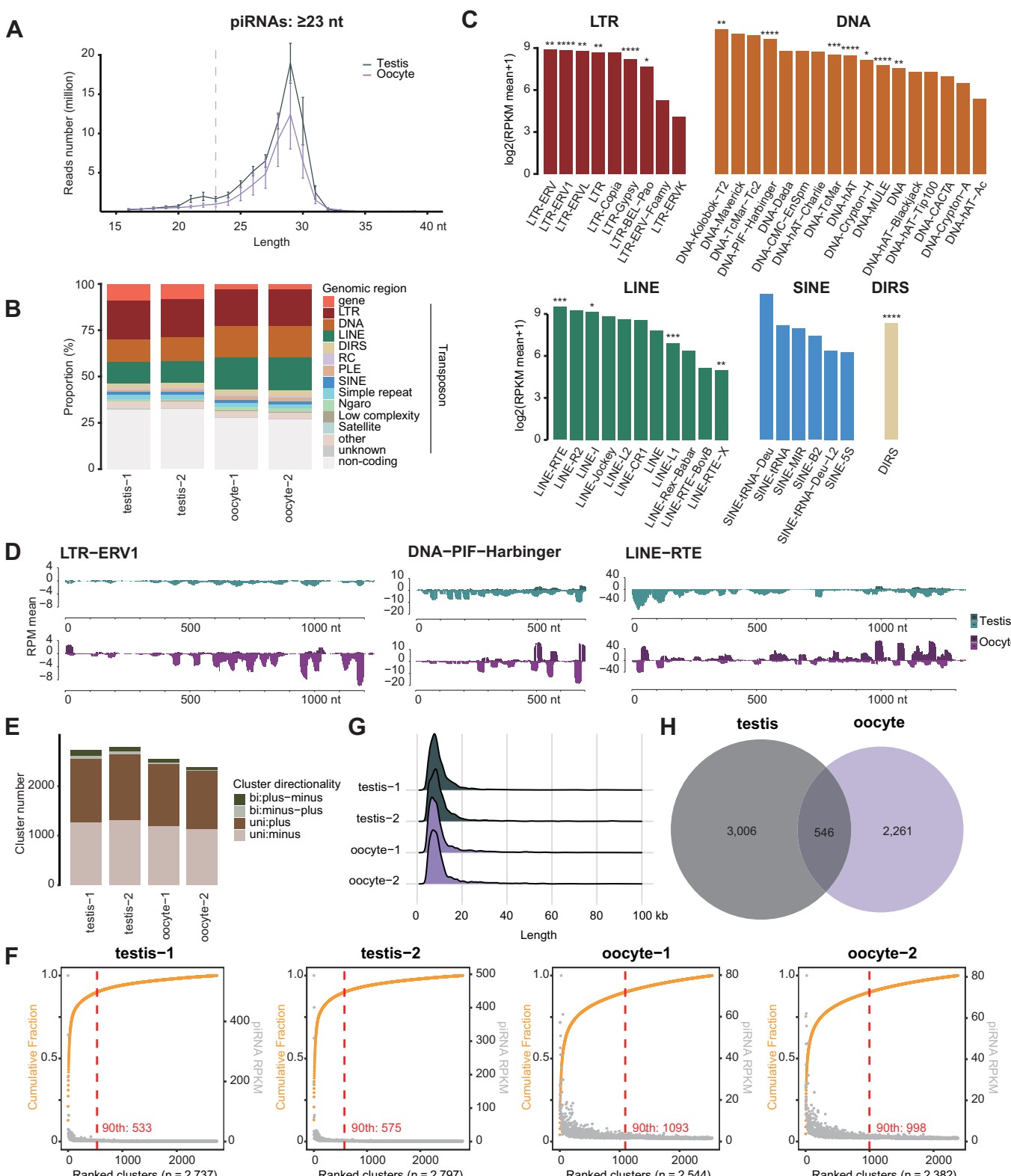

**Figure 4.  The axolotl genome encodes a diverse repertoire of transposon-targeting piRNAs.**

(A) Nucleotide (nt) length distribution from small RNA libraries. Mean and S.E.M. are presented. (B) Classification of piRNAs mapped to the axolotl genome. (C) Mean piRNA signal level over four testis and oocyte samples for each transposon family. ****, active copy number >1000; ***, active copy number >100; **, active copy number >10; *, active copy number >0. (D) Tracks for piRNAs targeting both sense and antisense strands of representative transposon consensus sequences. (E) Number of annotated piRNA clusters in the genome. (F) piRNA clusters ranked by the cumulative fraction of piRNA RPMs (orange). The RPKM of piRNAs from each cluster is displayed (gray). The red dashed line indicates the number of top-ranked clusters that collectively account for 90% of cluster-derived piRNAs (90th percentile). (G) Length distribution of annotated piRNA clusters in the genome. (H) The number of piRNA clusters that overlap between the two genders. For all panels, testis, $n = 2$; oocyte, $n = 2$. Source data are available online for this figure.

## Discussion

Our findings reveal that the mammalian version of the piRNA pathway comprising both cytoplasmic and nuclear components arose early in tetrapod evolution. PNLDC1, a pre-piRNA 3′ trimming exonuclease, mediates precursor piRNA trimming in mammals (Zhang et al, 2017; Ding et al, 2017; Nagirnaja et al, 2021). The presence of this enzyme distinguishes mammalian piRNA biogenesis from more ancient animal pathways. For instance, in *Drosophila*, a different enzyme Nibbler is employed (Hayashi et al, 2016; Han et al, 2011; Liu et al, 2011). We first observe the presence of *Pnldc1* gene in Coelacanth and as such propose that mammalian-like piRNA biogenesis arose early in lobe-finned fish evolution. In contrast, the evolution of a mammalian-like nuclear piRNA pathway occurred later than the more conserved cytoplasmic one. PIWIL4, SPOCD1, and TEX15 are key components of the mammalian nuclear piRNA pathway which arose at different stages of fish evolution. However, we find that all key piRNA pathway components and the de novo methylation machinery first co-occur in axolotl. The presence of PIWI proteins and mammalian piRNA pathway factors in the axolotl genome, with gonad specific expression, underscores the conserved nature of this pathway from amphibians to mammals (Figs. 1B and 2A). Furthermore, both male and female axolotl germlines exhibit abundant transposon-derived piRNAs, with sex-specific profiles (Fig. 4B,C). These piRNAs effectively target active transposons, as evidenced by robust ping-pong amplification cycles (Fig. 6B–E). This indicates an operational cytoplasmic piRNA pathway and post-transcriptional transposon silencing in both axolotl sexes. Remarkably, our study also suggests the existence of a piRNA-directed DNA methylation mechanism targeting young transposons in axolotl, a mechanism previously reported exclusively in mammals. Homologs of all key PIWIL4- and SPOCD1-associated nuclear factors identified in mammals are present in the axolotl genome and expressed in gonads (Fig. 2A). In addition, we observed that active transposons in the axolotl male germ cells are heavily DNA methylated (Fig. 7C,D). CGIs in axolotl are mostly associated with transposons (Fig. 7F), and in contrast to mammals, CGIs are heavily methylated in axolotl spermatozoa (Fig. 7E). Collectively, we provide evidence for a role of the nuclear piRNA pathway and DNA methylation in axolotl transposon control. However, genetic studies will be required to formally demonstrate a functional role for piRNA-directed transposon methylation in safeguarding the axolotl germline.

We also noted that not all tetrapods have a mammalian-like piRNA pathway (Fig. 1B). In frogs, several key nuclear factors are absent. While in birds, the crux of the nuclear pathway, such as PIWIL4, SPOCD1, C19ORF84, and TEX15, have been lost. This begs the question as to why these lineages have lost the nuclear branch of the

piRNA pathway. The answer may lie in the mechanism of germ cell acquisition. In tetrapods, the loss of the nuclear piRNA pathway correlates with preformation as the germline specification method that is used in birds and frogs (Extavour and Akam, 2003). While we do not understand the basis of the correlation between the presence of a nuclear piRNA pathway and the usage of epigenesis for germline specification in tetrapods, it might be linked to the evolution of epigenetic germline reprogramming. Preformation is mediated by germ plasm that is maternally inherited in the oocyte cytoplasm and instructs germ cell specification as one of the first embryonic cellular differentiation events (Mahowald, 1977). In this strategy, there is a continuity in the germline through the germ plasm, and maternally deposited piRNAs may confer continuous anti-transposon immunity as is the case in *Drosophila* (Brennecke et al, 2008). Both *C. elegans* and *Drosophila* have nuclear piRNA pathways and utilize preformation (Shirayama et al, 2012; Ashe et al, 2012; Luteijn et al, 2012; Rangan et al, 2011; Wang and Elgin, 2011; Klenov et al, 2011; Sienski et al, 2012; Extavour and Akam, 2003). The nuclear pathway in *Drosophila* is active in the germline and germline-associated somatic cells, where it prevents endogenous retroviruses that have the capacity to form infectious particles from attacking the germline (Sarot et al, 2004; Li et al, 2009a; Malone et al, 2009). This brings us to the curious case of lungfishes, which is similar to the axolotl, employs epigenesis for germline specification and possesses gigantic genomes with high transposon content (Nowoshilow et al, 2018; Wang et al, 2021; Schartl et al, 2024). However, while there is no evidence showing the axolotl genome size is currently growing (Wang et al, 2021; Schartl et al, 2024), lungfish genomes continue to expand, driven by young transposon activity (Schartl et al, 2024; Wang et al, 2021). Indeed, a recent study revealed that South American and African lungfish express few canonically sized piRNAs, leading to deficiencies in transposon control and contributing to massive genome expansion (Schartl et al, 2024). Consistent with this, we find that the piRNA pathway in lungfish lacks many genes involved in both the cytoplasmic and nuclear branches. Notably, among the sequenced lungfish, the South American lungfish has the largest genome (91 Gb) with 92.5% transposon content, and is currently expanding (Schartl et al, 2024). This further correlates with the fact that the South American lungfish has lost more piRNA pathway genes than the other lungfish species (Fig. 1B). The most significant difference between the lungfish and the axolotl is that the axolotl possesses a complete mammalian-like piRNA pathway in both the cytoplasm and nucleus. We present evidence indicating that both branches are active in axolotl, suggesting that stringent transposon control is in place. Thus, the current large size of the axolotl genome (32 Gb) (Nowoshilow et al, 2018) could reflect rapid transposon expansion in the past or a gradual expansion due to the number of remaining active copies combined with poor genome contraction processes (Sun et al, 2012).

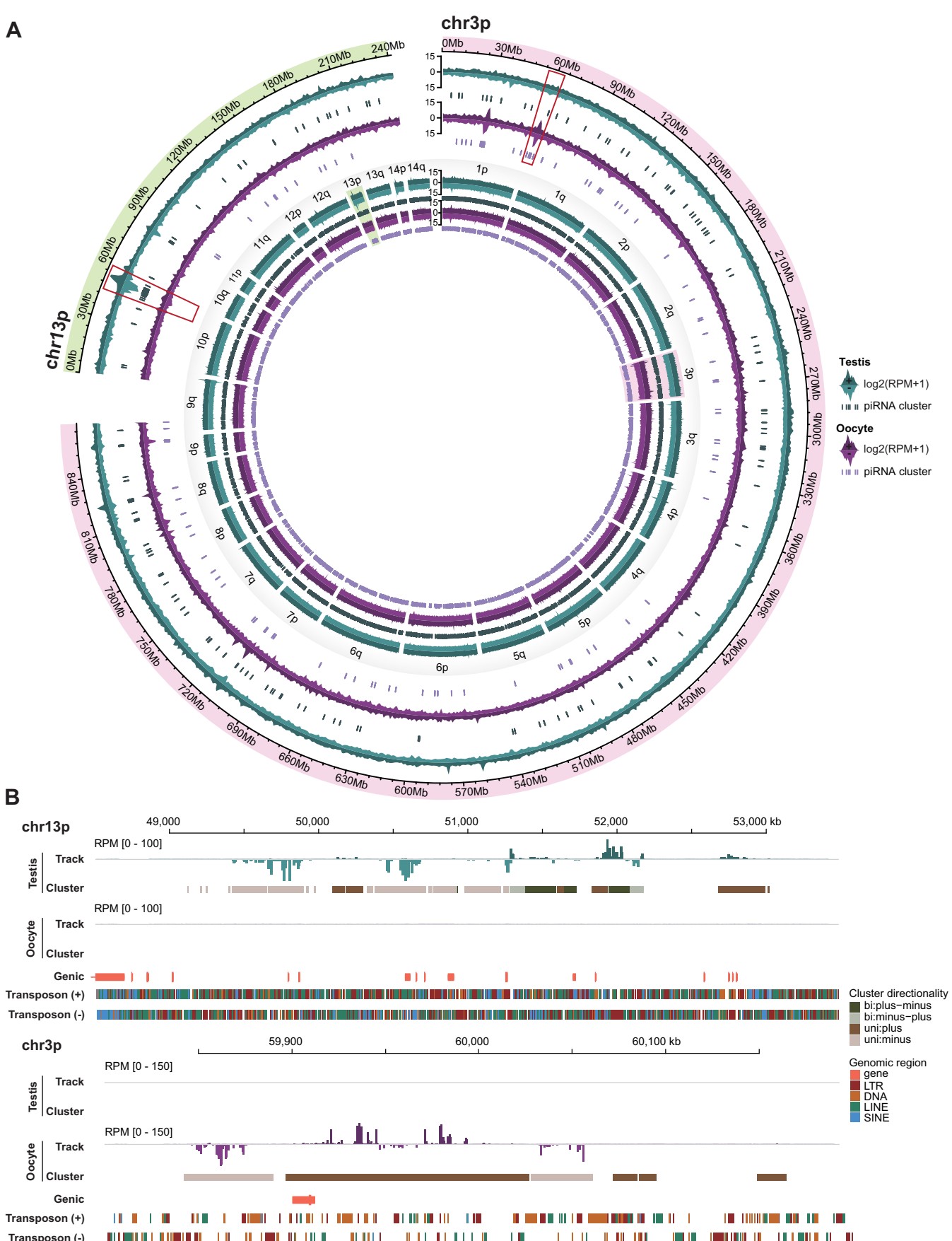

**Figure 5. The axolotl piRNA clusters exhibit sex-specific expression patterns.**

(A) Genome-wise distribution of piRNA signal and piRNA clusters (inner circle), with zoom-in view for chr3p and chr13p (outer circle). (B) Tracks on both sense and antisense strands for piRNA clusters are highlighted in (A). Genic regions and transposon regions are shown. For all panels, testis, $n = 2$; oocyte, $n = 2$. Source data are available online for this figure.

Tracing back to earlier evolutionary events, the emergence of lobe-finned fish, the ancestors of modern lungfish, marked the beginning of the water-to-land transition in vertebrates ~390 million years ago (Narkiewicz and Narkiewicz, 2015). These ancestral species possessed robust fins with skeletal structures that offered support and movement potential, but they remained fully aquatic. Remarkably, a pivotal evolutionary step occurred when a group of early semiaquatic animals emerged from lobe-finned fish, which were the ancient tetrapods around 360 million years ago (Coates et al, 2008). They finally became the common ancestors of modern amphibians, reptiles, birds, and mammals. Our findings suggest that the piRNA pathway underwent a refinement during the water-to-land transition. In lungfish, the crux of the modern piRNA pathway is present but flawed through gene loss, while the axolotl demonstrates a functional and seemingly efficient piRNA pathway, capable of both post-transcriptional and epigenetic silencing of transposons. This suggests that the piRNA pathway in mammals might trace its origins to a conserved mechanism present in ancient tetrapods. The evolution of these efficient piRNA-mediated transposon restriction mechanisms, especially piRNA-mediated DNA methylation, was pivotal to the evolution of germline reprogramming and genomic imprinting. Without the possibility to protect a hypomethylated genome and to precisely reinstall transposon methylation, germline reprogramming would be non-permissive. Thus, the emergence of an efficient piRNA pathway halted rapid or sustained transposon-mediated genome expansion to enable genome contraction, carving the compact terrestrial tetrapod genomes observed today.

# Methods

### Reagents and tools table

| Reagent/resource | Reference or source | Identifier or catalog number |
| --- | --- | --- |
| **Experimental models** | | |
| White axolotl strain (d/d) | Elly M. Tanaka Lab | DD151 |
| **Recombinant DNA** | | |
| **Antibodies** | | |
| **Oligonucleotides and other sequence-based reagents** | | |
| **Chemicals, enzymes, and other reagents** | | |
| TRIzol | Thermo Fisher | 15596026 |
| T4 RNA Ligase 2 truncated KQ | New England Biolabs | M0373L |
| T4 RNA Ligase 1 | New England Biolabs | M0204L |
| SuperScript II | Thermo Fisher | 18064014 |
| Kapa polymerase | Kapa Biosystems | KK2602 |

| Reagent/resource | Reference or source | Identifier or catalog number |
| --- | --- | --- |
| Pmel | New England Biolabs | R0560L |
| Monarch HMW DNA Extraction Kit | New England Biolabs | T3050L |
| NEBnext Enzymatic Methyl-seq kit | New England Biolabs | E7120L |
| **Software** | | |
| FastQC v0.11.8 | Babraham Bioinformatics | |
| Cutadapt v2.9 | https://github.com/marcelm/cutadapt/ | |
| Trim Galore v0.6.7 | Babraham Bioinformatics | |
| STAR v2.7.0e | https://github.com/alexdobin/STAR/ | |
| Bowtie2 v2.4.2 | http://bowtie-bio.sourceforge.net/bowtie2/index.shtml | |
| SAMtools v1.9 | http://www.htslib.org/ | |
| FeatureCounts v2.0.0 | http://subread.sourceforge.net/ | |
| RepeatMasker v4.1.5 | http://www.repeatmasker.org/ | |
| RepeatModeler v 2.0.5 | https://github.com/Dfam-consortium/RepeatModeler/ | |
| DeepTE | https://github.com/LiLabAtVT/DeepTE/ | |
| proTRAC v 2.4.3 | https://sourceforge.net/projects/protrac/ | |
| circlize | https://github.com/jokergoo/circlize | |
| gCluster | https://github.com/Xiangyang1984/Gcluster/ | |
| Bismark v0.24.0 | https://www.bioinformatics.babraham.ac.uk/projects/bismark/ | |
| SeqMonk v1.48.1 | Babraham Bioinformatics | |
| ggplot2 v3.3.5 | https://sourceforge.net/projects/ggplot2.mirror/files/v3.3.5/ | |
| RStudio v2022.07.0 | https://posit.co/download/rstudio-desktop/ | |
| R v3.5.1 | https://www.r-project.org/ | |
| Igv v2.3.72 g | http://software.broadinstitute.org/software/igv/ | |
| Illustrator software | Adobe Inc. | |
| **Other** | | |
| Illumina HiSeq2500 | Illumina | |
| Illumina NextSeq2000 | Illumina | |

## Axolotl strains and sample collection

White axolotl strain (d/d) was used for sample collection. Axolotl husbandry was described in detail previously (Khattak et al, 2014). All lines were bred and maintained at the Research Institute of Molecular Pathology (IMP) facilities. Handling and surgical procedures adhered to local ethics committee guidelines, and animal experiments were conducted with approval from the Magistrate of Vienna (Genetically Modified Organism Office and MA58, City of Vienna, Austria, license GZ51072/2019/16 and license GZ665226/2019/21).

Small RNA-seq was performed on axolotl testis and oocyte samples, each with two biological replicates. EM-seq was performed on axolotl spermatozoa with three biological replicates. Samples were collected from the adult axolotl with hCG (human chorionic gonadotropin) injection to induce gamete maturation.

## Small RNA sequencing for axolotl germline

RNA was extracted from two testis and two oocyte samples of the d/d white axolotl strain using TRIzol reagent (15596026), following the manufacturer's protocol. Small RNA libraries were prepared as described in a previous publication (Jayaprakash et al, 2011). Briefly, 15 µg of total RNA was spiked with radioactively labeled marker RNAs of 19 and 35 nucleotides in length and resolved on a denaturing polyacrylamide gel. Small RNAs were excised from the gel and subjected to 3′ linker ligation using pre-adenylated DNA linkers containing four random nucleotides at the 5′ end to minimize ligation bias, following established protocols (Jayaprakash et al, 2011). The ligation was performed using T4 RNA Ligase 2 truncated KQ (M0373L).

The ligation products were purified on a denaturing polyacrylamide gel, excised based on the shifted spike-in signal, and ligated to RNA 5′ adapters containing four random nucleotides using T4 RNA Ligase 1 (M0204L). After a final purification step via denaturing polyacrylamide gel, the small RNA libraries were completed by reverse transcription using SuperScript II (18064014), followed by PCR amplification with Kapa polymerase (KK2602). The spike-in RNAs were digested using PmeI (R0560L), and the libraries were further purified using low-melt agarose gel excision. Sequencing was performed on an Illumina HiSeq2500 platform in 50-bp single-end read mode.

## Whole-genome methylation sequencing for axolotl spermatozoa

DNA was extracted from three sperm samples of the white axolotl strain using Monarch HMW DNA Extraction Kit (NEB, T3050L) and solubilized in 10 mM Tris, pH 9.0, 0.5 mM EDTA. Methyl-seq libraries were prepared using the NEBnext Enzymatic Methyl-seq kit (NEB, E7120L) according to the manufacturer's instructions. Sequencing was performed on an Illumina NextSeq2000 platform in 150-bp paired-end read mode.

## Reference genome for bioinformatics analysis

The axolotl reference genome AmexG_v6.0 from https://www.axolotl-omics.org was employed (Nowoshilow et al, 2018). Axolotl gene annotation AmexT_v47 from https://www.axolotl-omics.org was employed (Nowoshilow et al, 2018). De novo transposon annotation was obtained using custom pipeline (see "Axolotl transposon annotation").

## Phylogeny analysis for piRNA pathway and germline specification mechanism in vertebrates

A phylogenetic tree was constructed according to the UCSC Genome Browser (Perez et al, 2024) (https://genome.ucsc.edu/). Icons for vertebrate species were obtained from PhyloPic (https://www.phylopic.org/), and all icons have been dedicated to the public domain.

Genome size and transposon percentage were obtained from UCSC assembly hubs (Perez et al, 2024) (https://hgdownload.soe.ucsc.edu/hubs/vertebrate/asmStats.html). For lungfish, genome size and transposon percentage were from Schartl et al (Schartl et al, 2024). Germline specification mechanisms for vertebrate species were collected from Extavour et al (Extavour and Akam, 2003) and Hansen et al (Hansen and Pelegri, 2021).

For piRNA pathway-related gene annotation, genome annotated files were obtained from UCSC Genome Browser (Perez et al, 2024) (https://genome.ucsc.edu/), Ensembl Gene Annotation (Howe et al, 2021) (https://www.ensembl.org/) or NCBI Gene Database (Sayers et al, 2022) (https://www.ncbi.nlm.nih.gov/datasets/gene/). For axolotl and lungfishes, genome annotated files were from Nowoshilow et al (Nowoshilow et al, 2018) and Schartl et al (Schartl et al, 2024), respectively. To identify potentially unannotated genes, additional searches were performed using NCBI tBLASTn against the reference genomes. For details, please refer to Dataset EV1.

## RNA-seq analysis for axolotl tissues

RNA-seq datasets for axolotl testis, ovary, and 16 somatic tissues were obtained from previously published datasets GSE92429 (Bryant et al, 2017) and GSE182746 (Ye et al, 2022). FastQC v0.11.8 (Andrews, 2010) was applied for raw reads quality control. Alignment was performed using STAR v2.7.0e (Dobin et al, 2013), with no more than three mismatches (--outFilterMismatchNmax 3). The output BAM files were sorted and indexed with SAMtools v1.9 (Li et al, 2009b) for downstream analysis. FeatureCounts v2.0.0 (Liao et al, 2014) was used to quantify gene expression by exon counts (-g gene_id -t exon), and only the unique mapped reads were used for quantification. Read counts were normalized to RPKM for visualization. Heatmap was plotted by ggplot2 v3.3.5 (Wickham, 2016) in RStudio (RStudio Team, 2020).

## Axolotl transposon annotation

Axolotl transposon annotation was constructed using a multi-step pipeline by combining the standard RepeatMasker annotation workflow (Smit et al, 2013) with the deep-learning-based annotation tool DeepTE (Yan et al, 2020).

First, transposon consensus sequences were generated. Transposon ancestor sequences were retrieved from the Dfam database using famdb.py from RepeatMasker v4.1.5 (Smit et al, 2013) (--ancestors). De novo transposon consensus sequences were identified with RepeatModeler v2.0.5 (Flynn et al, 2020) employing RMBlast as the search engine (-engine rmblast) and incorporating the LTR structural discovery pipeline alongside the RepeatScout/

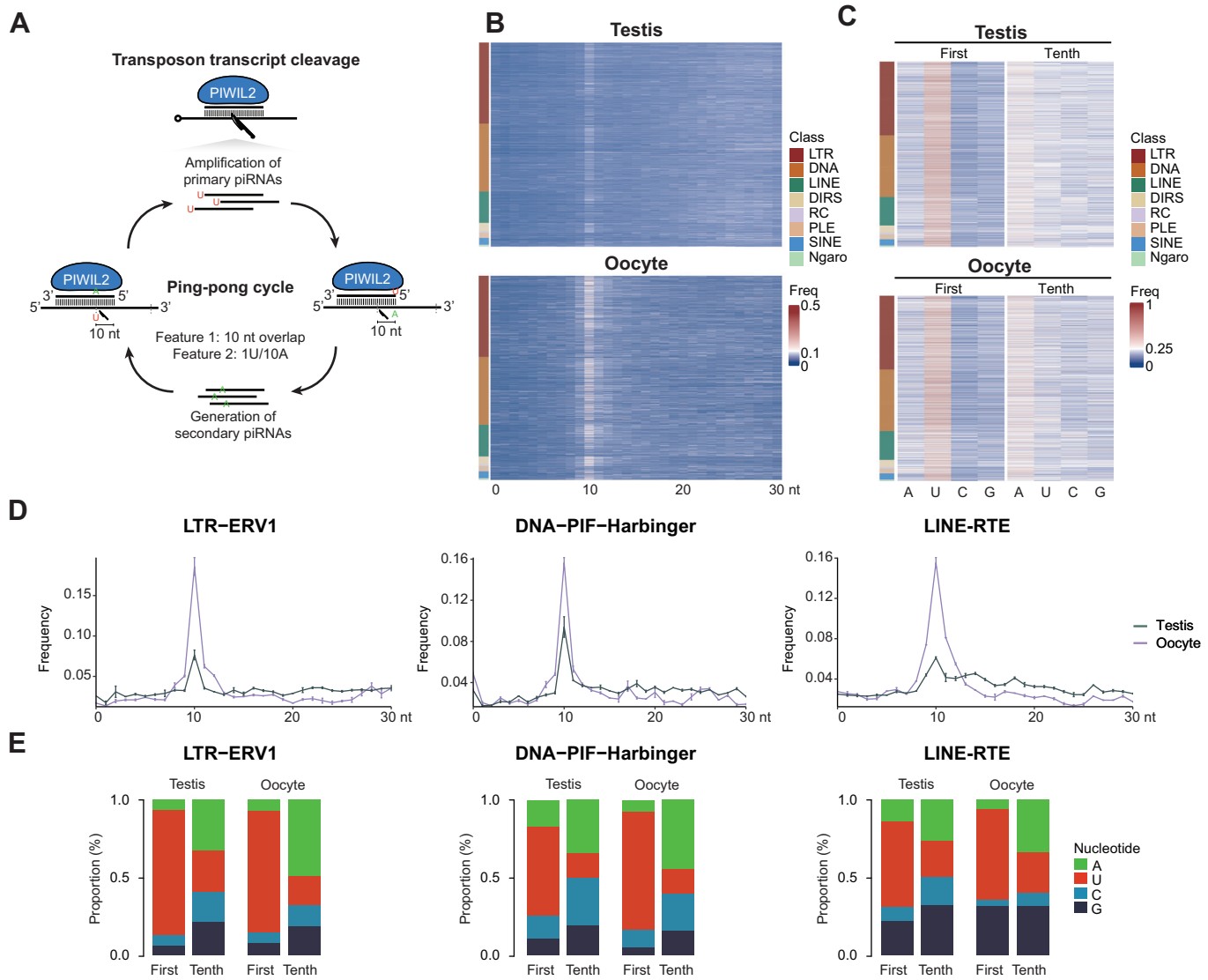

**Figure 6. piRNA pathway post-transcriptionally silences transposons in both male and female germlines.**

(A) Schematic of the ping-pong cycle and transposon silencing. Two features of piRNAs generated from the ping-pong cycle for transposon transcript cleavage are displayed. One is the 10 nt overlap between 5′ of complementary piRNA pairs, the other is the 1U/10A preference. (B, D) Relative frequency of the nucleotide distance between 5′ of complementary piRNA pairs. 5′ nucleotide overlap frequency over all families from major transposon classes (B) and representative transposon families (D) are shown. Mean and S.E.M. are presented in (D). (C, E) Nucleotide composition of the first and tenth position in piRNAs. Nucleotide proportion over all families from major transposon classes (C) and representative transposon families (E) are shown. A adenine, U uracil, C cytosine, G guanine. For all panels, testis, *n* = 2; oocyte, *n* = 2. Source data are available online for this figure.

RECON pipeline (-LTRStruct). For unclassified transposon consensus sequences, DeepTE was used to further categorize them into Metazoan transposon classes (-sp M). The three consensus libraries were combined for genome annotation with RepeatMasker.

Then the genome was annotated in three iterative rounds using RepeatMasker v4.1.5. In the first round, simple repeats were annotated and masked (-e rmblast -noint -xsmall). For the second round, known transposons were annotated using the combined consensus library and applied to the simple repeats masked genome (-e rmblast -nolow), with the results masked prior to the third round. In the third round, the still unclassified transposons from DeepTE were annotated (-e rmblast -nolow). Finally, the outputs

from all three rounds were merged to generate a comprehensive transposon annotation for the axolotl genome.

## Potentially active transposon identification

For axolotl, the de novo transposon annotation file was used for potentially active transposon identification. Potentially active transposon copies were identified using two criteria. First, copies were required to have lengths close to the full-length consensus sequence, excluding truncated elements. Length thresholds were determined according to published references for each transposon family (Dataset EV2). Second, only young copies with less than 5%

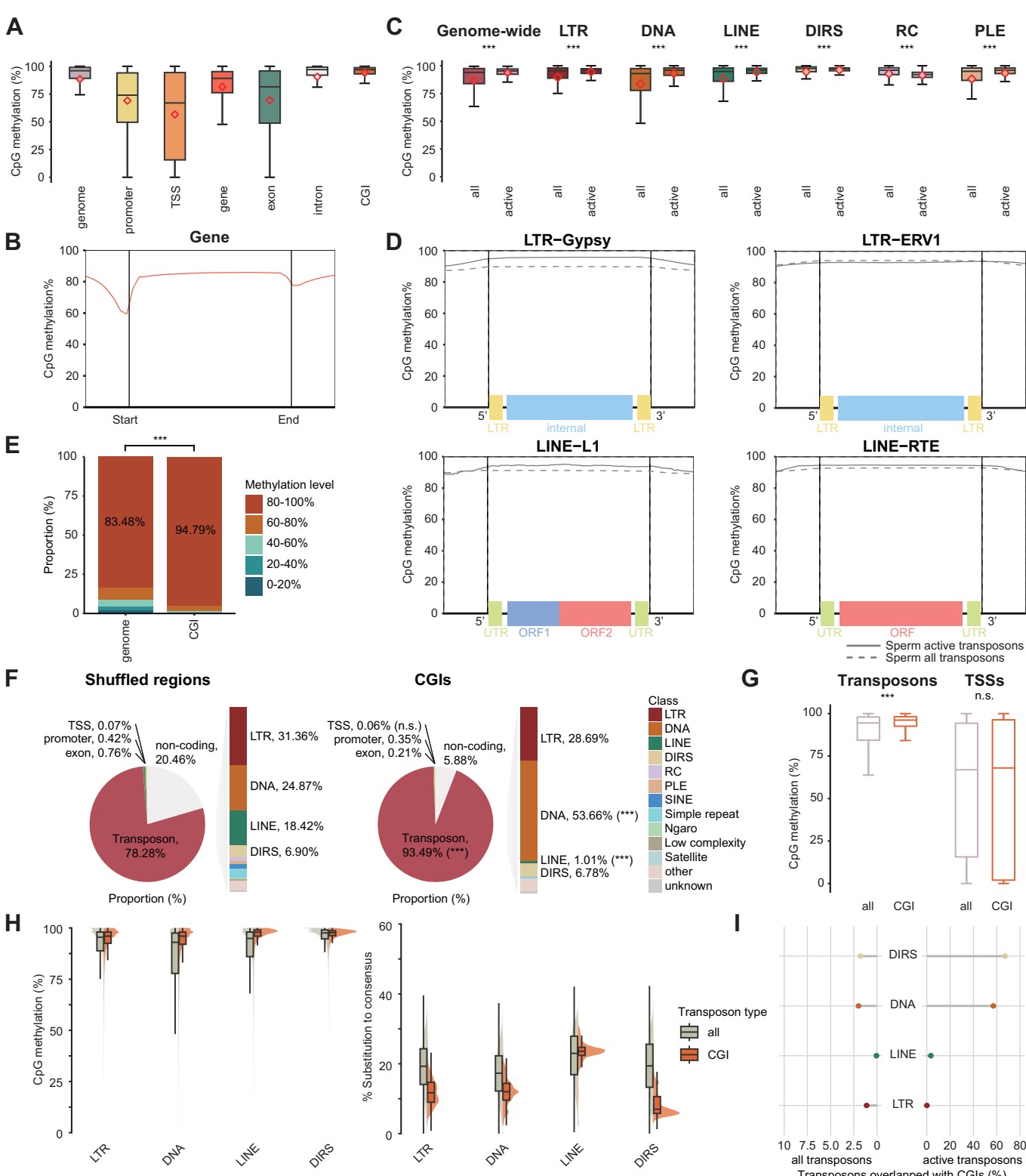

sequence divergence from the consensus (percDiv) were included, as higher divergence indicates older copies with accumulated mutations (Novick et al, 2011; Zoch et al, 2020). Transposon copies meeting both criteria were classified as potentially active transposons (Dataset EV2).

## Small RNA analysis

FastQC v0.11.8 (Andrews, 2010) was applied for raw reads quality control. The NEB adapter sequence (AGATCGGAAGA) was trimmed from the 3′ ends of raw FASTQ files using Cutadapt v2.9 (Martin,

**Figure 7.   The genome of axolotl spermatozoa is highly methylated.**

(A) Percentages of CpG methylation levels over the whole genome or certain genomic elements. (B) Metaplots of mean CpG methylation over gene bodies and adjacent 2 kb. (C) Percentages of CpG methylation levels over all genomic transposons or certain transposon classes. Methylation levels over all copies or only active copies are shown. (D) Metaplots of mean CpG methylation for selected transposon families over all transposon/active transposon bodies and adjacent 2 kb. (E) Distribution of CpG methylation levels across the whole genome or CGI regions. (F) Genomic annotation for CGIs or shuffled regions. (G) CpG methylation levels over all transposon/TSS regions or regions overlapped with CGIs. (H) Distribution of CpG methylation level and substitution rate for all transposon copies or copies overlapped with CGIs. (I) Percentages of transposons overlapped with CGIs. For boxplots, the middle line represents the median; boxes represent the 25th (bottom) and 75th (top) percentiles; whiskers represent median ± 1.5× interquartile range; and outside values are not shown. Rhombus, mean level. For statistical tests, \*\*\*P value <0.001, n.s. not significant; (E, F) Chi-square test, (C, G) unpaired two-sided *t* test. The exact P values for (C, E–G) are provided in Source Data. For all panels, spermatozoa, *n* = 3; pooled for analysis. Source data are available online for this figure.

2011). Reads shorter than 16 bases after trimming were discarded (-m 16), as well as those lacking adapters (--trimmed-only). Four random nucleotides were clipped off from each end. Small RNA sequences with a minimum length of 23 bases were defined as piRNAs for downstream analysis to exclude miRNAs and siRNAs (Dataset EV3). For alignment, reads were mapped to the axolotl genome or transposon consensus sequences with STAR v2.7.0e (Dobin et al, 2013) with the following parameters: up to 1000 multimapped reads permitted (--outFilterMultimapNmax 1000), a maximum of three mismatches allowed (--outFilterMismatchNmax 3), and a minimum of 15 matched bases required (--outFilterMatchNmin 15). All multimapped reads were exported (--outSAMmultNmax -1) and were included for quantification over genic or transposon regions, with each read contributing 1/n to the counts. For genomic annotation, mapped reads were annotated based on the axolotl gene and transposon annotations. Reads that did not align to any genomic features were classified as 'non-coding'.

## piRNA cluster annotation

piRNA clusters were detected using the proTRAC v2.4.3 (Rosenkranz and Zischler, 2012) standard pipeline (Dataset EV4). First, redundant sequences and low-complexity reads were removed with TBr2_collapse.pl and TBr2_duster.pl with default parameters, respectively. The processed reads were then aligned to the axolotl genome using sRNAmapper.pl, allowing up to three mismatches, with a minimum of 15 matched bases (-mismatch 3 -seedmatch 15). Finally, piRNA cluster identification was conducted with proTRAC_2.4.3.pl, using a sliding window size of 5000, a piRNA density threshold of 0.01, and restricting piRNA lengths to 23–33 nucleotides (-swsize 5000 -pdens 0.01 -pimin 23 -pimax 33). piRNA clusters and signals were visualized using the circlize R package (Gu et al, 2014). piRNA signals were aggregated in 1 Mb genomic bins by summing 1/n-weighted scores of all mapped piRNAs, where n is the number of mapping sites per piRNA. The aggregated signals were then normalized to RPM and log2-transformed for visualization.

## Ping-pong cycle analysis

Ping-pong analysis was performed using only reads mapped to transposons. For each overlapping sense and antisense read pair, the distance between their 5′ ends was calculated, and the counts were presented as relative frequencies. The nucleotide composition at the first and tenth positions of piRNAs was analyzed and expressed as proportions. Heatmap, line charts, and bar plots were generated using ggplot2 v3.3.5 (Wickham, 2016) in RStudio (RStudio Team, 2020).

## CpG island annotation

CpG islands (CGIs) in axolotl were annotated by gCluster (Li et al, 2020) with the canonical Gardiner-Garden and Frommer criteria (Gardiner-Garden and Frommer, 1987). Genomic regions with CG contexts were extracted (pattern = CG), and were then filtered to define CGIs based on G + C content (≥50%), length (≥200 bp), observed/expected CpG ratio (≥0.6), and P value (≤1e-5). A total of 248,988 CGIs could be identified in the axolotl genome.

## Whole-genome methylation analysis

For axolotl spermatozoa, three biological replicate samples were prepared and sequenced as described in the "Whole-genome methylation sequencing for axolotl spermatozoa" under "Methods". To increase sequencing coverage and depth, the three replicates were merged for downstream analysis.

Raw sequencing reads were assessed for quality using FastQC v0.11.8 (Andrews, 2010) and processed using Trim Galore v0.6.7 (Krueger, 2019) with Cutadapt v3.4 (Martin, 2011) to remove adapter sequences and low-quality bases (--trim-n --clip_R2 5 --quality 20 --length 25). Trimmed paired-end reads were aligned to the reference genome or transposon consensus with the standard pipeline from Bismark v0.24.0 (Krueger and Andrews, 2011) with Bowtie2 v2.4.2 (Langmead and Salzberg, 2012), allowing up to 1 bp mismatch and permitting multimapping, with one random alignment reported for ambiguous mappings (-N 1 --ambig_bam). The mapped reads were deduplicated by deduplicate_bismark. CpG methylation data were then extracted using bismark_methylation_extractor with default parameters. For axolotl spermatozoa samples, the methylation conversion rate was determined by aligning all reads to unmethylated lambda and methylated pUC19 spike-in controls using the same Bismark pipeline (Dataset EV5).

The methylation coverage data generated by Bismark v0.24.0 (Krueger and Andrews, 2011) were imported into SeqMonk v1.48.1 (Andrews, 2022) for quantification and visualization. Methylation analysis was performed using a 50-CpG running window approach (Positions per window = 50), retaining only probes with at least 10 read counts (Minimum observations to include feature = 10). For each probe, the mean methylation level of the read counts was calculated. CpG methylation levels over genomic features were determined as the mean methylation of probes overlapping the respective features.

For genomic features, transcription start sites (TSS) were the first base of annotated transcripts, promoters were defined as the 2 kb regions upstream of the TSS, and genes were divided into exon and intron regions based on gene annotations. All transposon

copies and potentially active transposon copies were identified as described (see "Axolotl transposon annotation and Potentially active transposon identification"). Transposon methylation analysis in this study excluded repeats overlapping exonic regions. Boxplots, scatterplots and correlation analysis were generated using ggplot2 v3.3.5 (Wickham, 2016) in RStudio (RStudio Team, 2020). Statistical significance was assessed using unpaired two-sided *t* test.

## Statistics and reproducibility

Sample size was not predetermined using statistical methods, and no data were excluded from analyses. Allocation and outcome assessments were not blinded. Statistical analyses were performed in R v3.5.1 (R Core Team, 2018), with specific test methods detailed in the figure legends.

## Data availability

All sRNA-seq and EM-seq data have been deposited at the GEO database under the accession number GSE290049. Scripts used for transposon annotation, RNA-seq, sRNA-seq and EM-seq analyses and plots are available on github (https://github.com/XinyuXiang/AxolotlpiRNApathway/).

The source data of this paper are collected in the following database record: biostudies:S-SCDT-10_1038-S44318-025-00631-w.

## Peer review information

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

## Acknowledgements

This research was supported by funding from Wellcome to D.O'C. (106144 and 225237), International Scientific Research Cooperation Cultivation Project of Zhejiang University International Campus to WL and DO'C, ZJU-UoE Joint Research Project funding from the ZJE institute to WL and DO'C, Natural Science Foundation of China to WL (32170551). We also sincerely acknowledge the funding from the Wellcome Centre for Cell Biology (203149 and multi-user equipment grants 108504 and 092076); the Wellcome Discovery Research Platform for Hidden Cell Biology (226791); and support from the bioinformatics cores of the Wellcome Discovery Research Platform for Hidden Cell Biology. We thank Dr. Yuka Kabayama and Dr. Rebecca V Berrens for their generous assistance on sRNA-seq and EM-seq analysis, and Dr. Julius Brennecke, Dr. Ansgar Zoch, David MacLeod and Tamoghna Chowdhury for their insightful discussions and help with manuscript review. We also extend our gratitude to staffs at the EMBL GeneCore facility in Heidelberg, Germany, for preparing the EM-seq libraries and sequencing all libraries.

## Author contributions

**Xinyu Xiang**: Resources; Data curation; Formal analysis; Validation; Investigation; Visualization; Methodology; Writing—original draft; Project administration; Writing—review and editing. **Anni Gao**: Formal analysis; Visualization; Methodology. **Dominik Handler**: Validation; Methodology; Writing—review and editing. **Francisco Falcon**: Validation; Methodology; Writing—review and editing. **Diego Rodriguez-Terrones**: Formal analysis; Investigation; Methodology; Writing—review and editing. **Sergej Nowoshilow**: Supervision; Visualization; Methodology; Writing—review and editing. **Wanlu Liu**: Supervision; Funding acquisition; Writing—review and editing. **Elly M Tanaka**: Resources; Supervision; Validation; Methodology; Project administration; Writing—review and editing. **Dónal O'Carroll**: Conceptualization; Resources; Supervision; Funding acquisition; Investigation; Writing—original draft; Project administration; Writing—review and editing.

Source data underlying figure panels in this paper may have individual authorship assigned. Where available, figure panel/source data authorship is listed in the following database record: biostudies:S-SCDT-10_1038-S44318-025-00631-w.

## Disclosure and competing interests statement

The authors declare no competing interests.

# Expanded View Figures

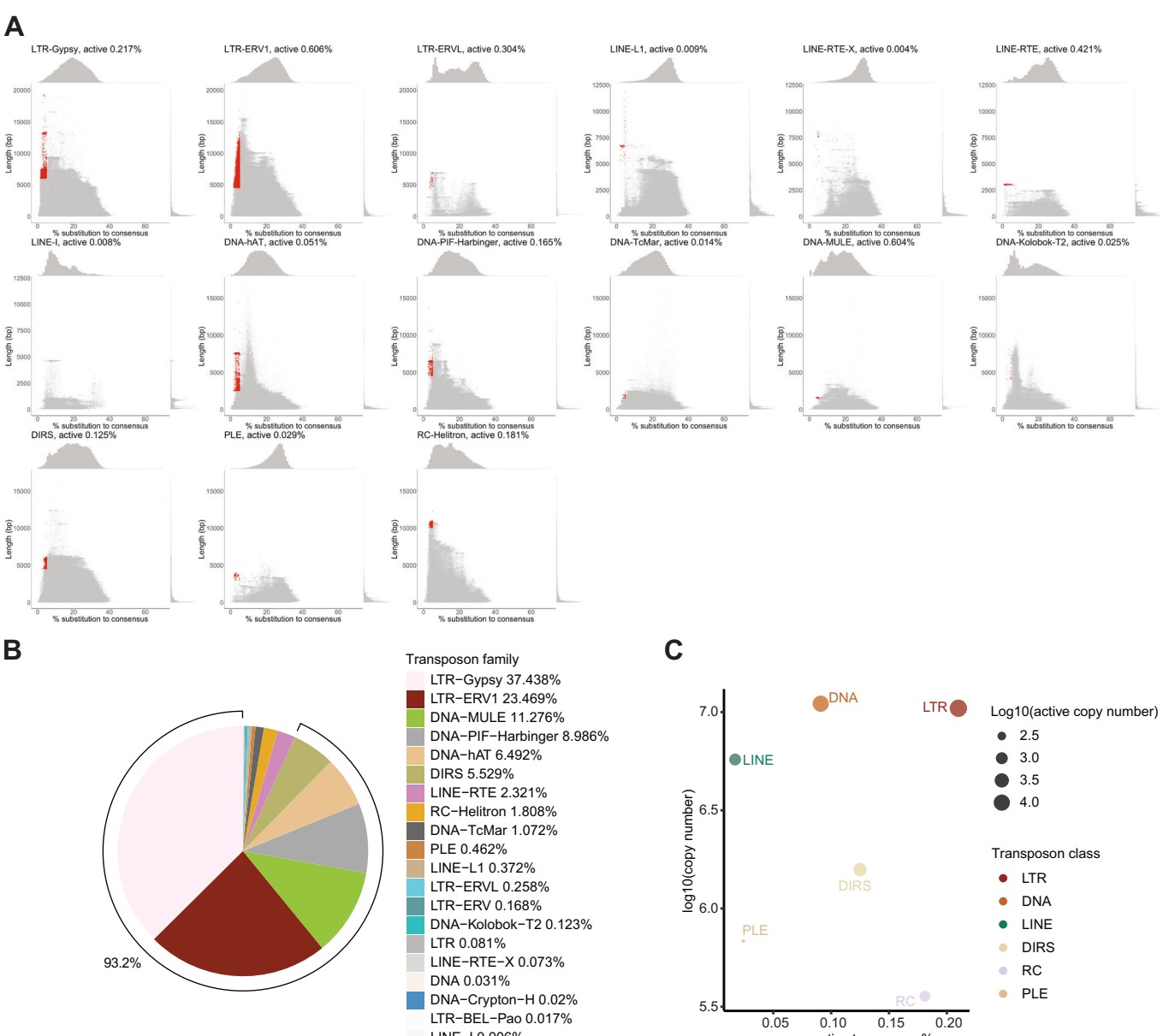

**Figure EV1.  Identification and distribution of potentially active transposons in the axolotl genome.**

(**A**) Distribution of substitution rate and sequence length for every single copy in each transposon family. Defined potentially active copies are highlighted in red. (**B**) Composition of potentially active copies. (**C**) Scatter plot of percentage of potentially active copy number and total copy number for transposon classes. Transposon classes without potentially active copies are omitted.

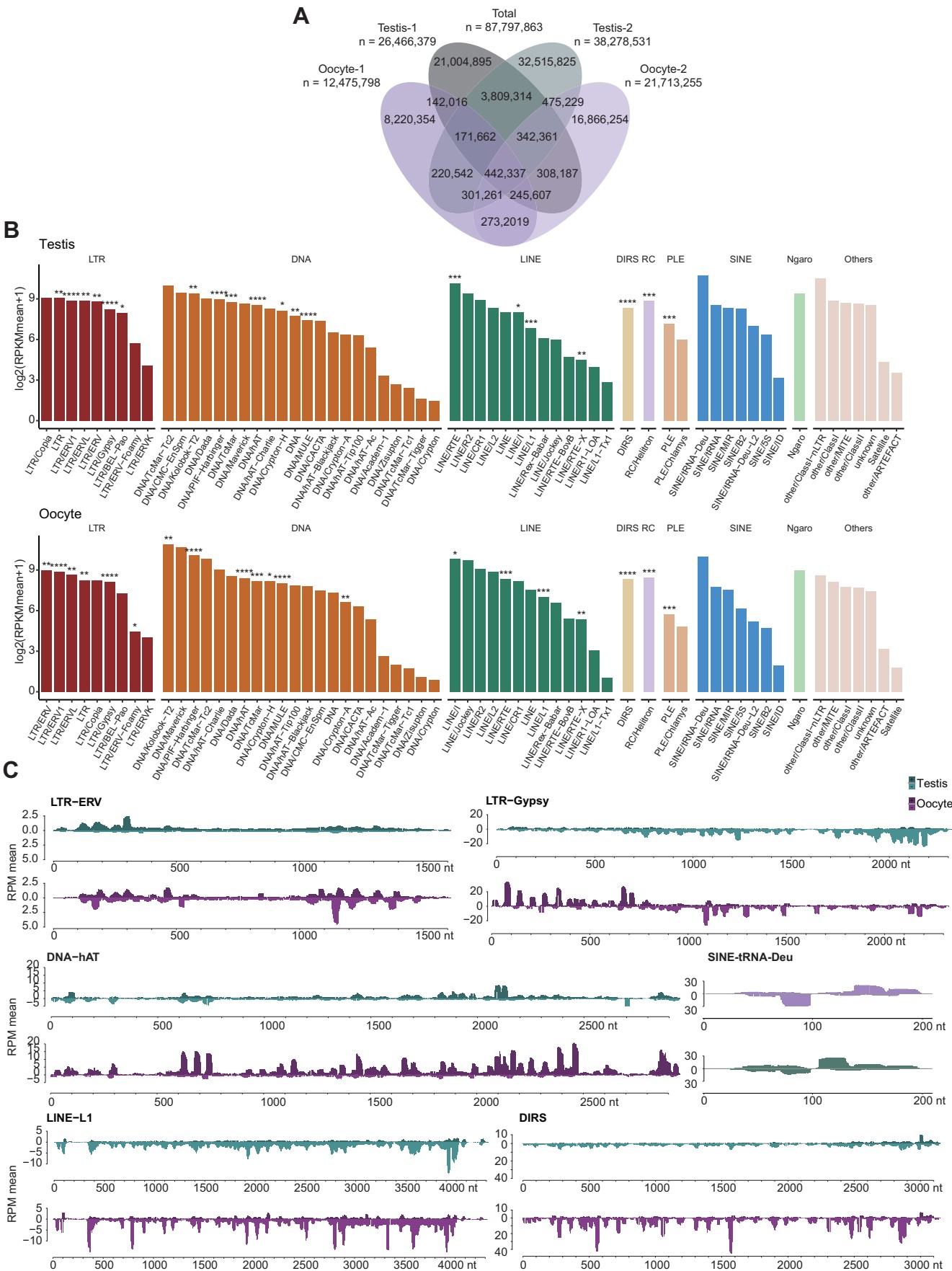

**Figure EV2.  piRNA repertoires and transposon targeting in axolotl germlines.**

(A) The overlap of unique piRNA sequences among the four samples. (B) Mean piRNA signal level over testis or oocyte for each transposon family. ****, active copy number >1000; ***, active copy number >100; **, active copy number >10; *, active copy number >0. (C) Tracks for piRNAs targeting both sense and antisense strands of representative transposon consensus sequence. For all panels, testis, $n = 2$; oocyte, $n = 2$.

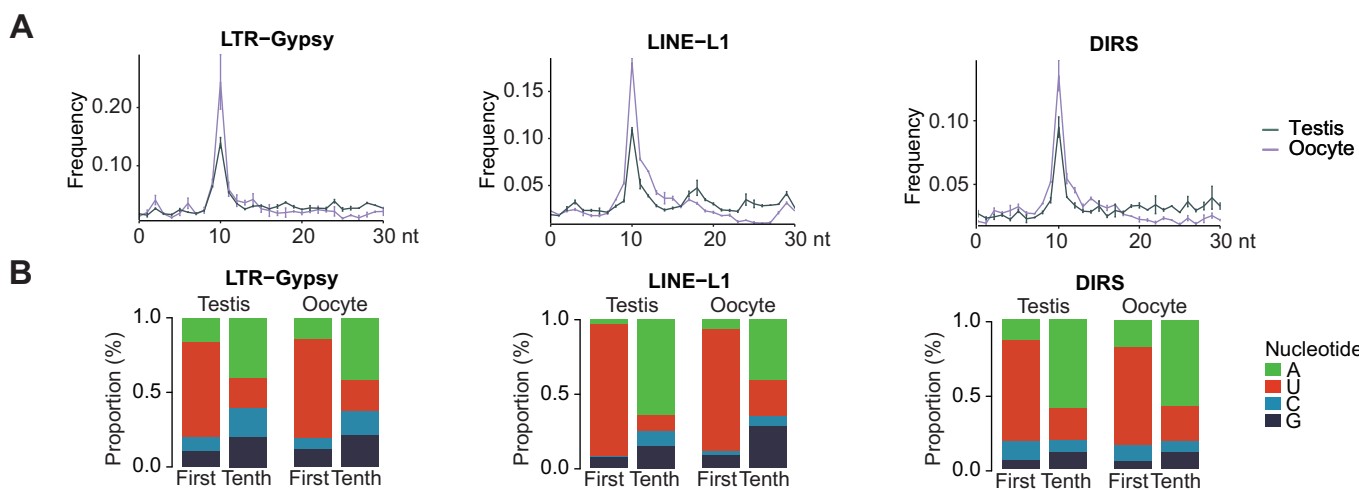

**Figure EV3. Features of ping-pong cycle in axolotl germlines.**

(A) Relative frequency of the nucleotide distance between 5' ends of complementary piRNA pairs over representative transposon families. Mean and S.E.M. are presented. (B) Nucleotide composition of the first and tenth position in piRNAs over representative transposon families. A, Adenine; U, Uracil; C, Cytosine; G, Guanine. For all panels, testis, $n = 2$; oocyte, $n = 2$.

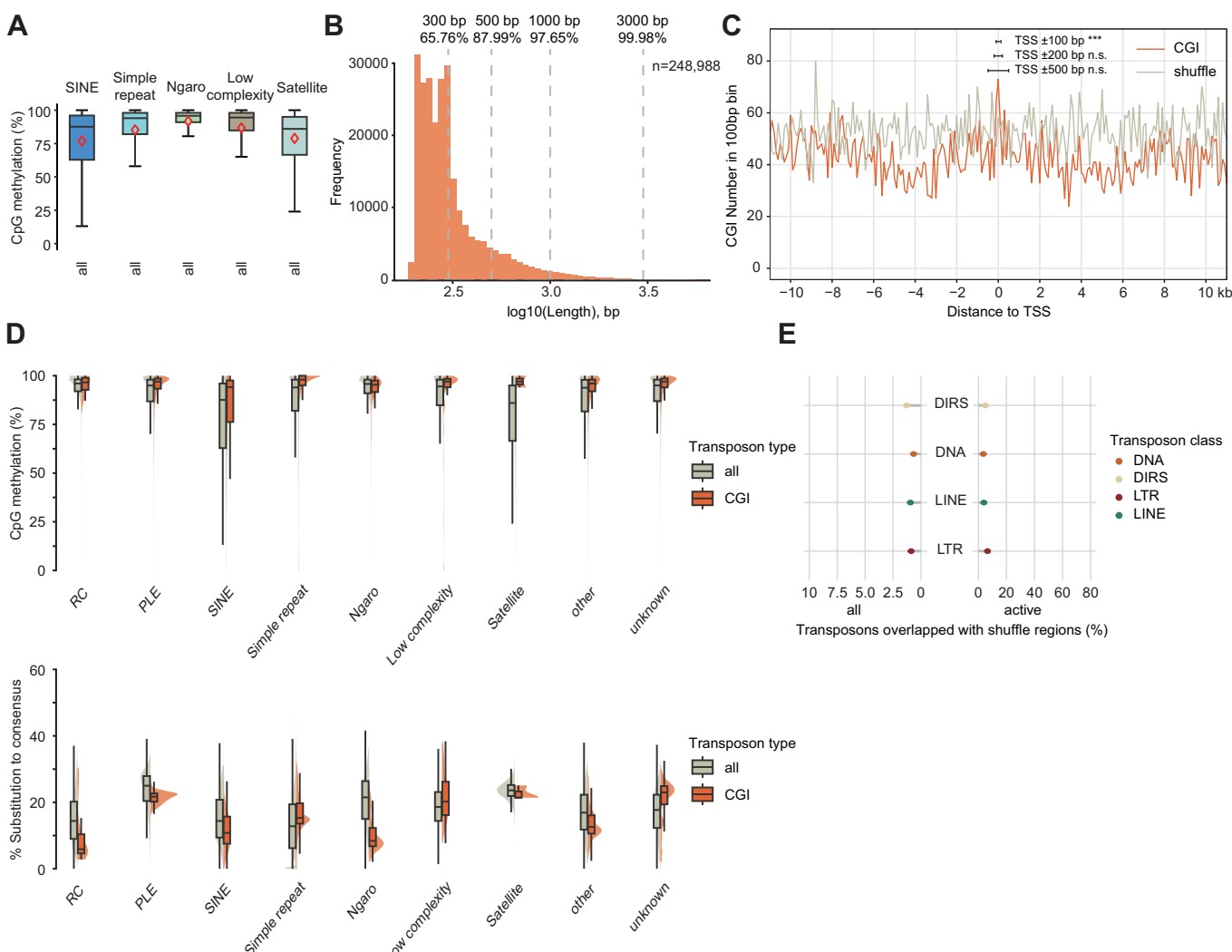

**Figure EV4. DNA methylation landscape of minor transposon classes and CpG islands in axolotl spermatozoa.**

(**A**) Percentages of CpG methylation levels over all copies of minor transposon classes in the axolotl. For boxplots, the middle line represents the median; boxes represent the 25th (bottom) and 75th (top) percentiles; whiskers represent median ± 1.5× interquartile range; and outside values are not shown. Rhombus, mean level. (**B**) Length distribution of all CGI regions ($n = 248{,}988$). (**C**) CGI enrichment over TSS and adjacent regions. For statistical tests, ***$P$ value < 0.001, n.s. not significant; Chi-square test. (**D**) Distribution of CpG methylation level and substitution rate for all transposon copies or copies overlapped with CGI of minor transposon classes. (**E**) Percentages of transposons overlapped with shuffled region. For all panels, spermatozoa, $n = 3$; pooled for analysis.

