## [Peer Review File · The EMBO Journal]

A mammalian-like piRNA pathway in Axolotl reveals the origins of piRNA-directed DNA methylation

Xinyu Xiang, Anni Gao, Dominik Handler, Francisco Falcon, Diego Rodriguez-Terrones, Sergej Nowoshilow, Wanlu Liu, Ely Tanaka, and Donal O'Carroll

Corresponding author: Donal O'Carroll (donal.ocarroll@ed.ac.uk)

Review Timeline:

Submission Date:	15th Apr 25
Editorial Decision:	17th May 25
Revision Received:	8th Aug 25
Editorial Decision:	1st Oct 25
Revision Received:	6th Oct 25
Accepted:	17th Oct 25

Editor: Yehu Moran

Transaction Report:

Dear Dr. O'Carroll,

Thank you for submitting your manuscript for consideration by the EMBO Journal. It has now been seen by three referees whose comments are shown below.

Given the referees' positive recommendations, I would like to invite you to submit a revised version of the manuscript, addressing the comments of all three reviewers. I should add that it is EMBO Journal policy to allow only a single round of revision, and acceptance of your manuscript will therefore depend on the completeness of your responses in this revised version.

Please pay special attention to the similar remarks by Referees #1 and #3 that concern the discrepancy between the title and major subject of the paper (i.e, evolution of the mammalian piRNA pathway vs. the piRNA pathway of axolotl). It would be important to resolve this issue in the revision.

We recommend that after consulting with your co-authors you will send us via email in the next few weeks a revision plan. This should cover any additional experiments and/or analyses you plan to include in your revision. Moreover, if some suggestions by the referees are impossible to perform due to technical limitations or expected to take too long (our standard revision time frame is 3 months) this would be a good opportunity to mention these challenges. From our experience sending us the revision plan in advance will help both you and us to set our expectations and can facilitate a smoother revision process.

Thank you for the opportunity to consider your work for publication. I look forward to your revision.

Yours sincerely,

Yehu Moran
Academic Editor
The EMBO Journal

We realize that it is difficult to revise to a specific deadline. In the interest of protecting the conceptual advance provided by the work, we recommend a revision within 3 months (15th Aug 2025). Please discuss the revision progress ahead of this time with the editor if you require more time to complete the revisions.

Referee #1:

Summary

Mammalian early development features a genome-wide demethylation phase during epigenetic reprogramming, which renders the germline genome susceptible to the activation and transposition of transposable elements (TEs). The piRNA pathway plays a key role in counteracting this threat by suppressing TEs and preserving genome integrity.

In this work, "On the origin of the mammalian piRNA pathway", the authors investigate the genetic framework and expression of piRNA genome defense pathway components in the salamander axolotl, identifying homologous piRNA pathway components conserved with the mouse system as a proxy for the "modern" mammalian pathway. They annotate the transposable element landscape in this species and analyze piRNA pools along with their genomic source loci, as well as stereotypical piRNA amplification 'ping-pong' signatures. From this, they identify piRNAs targeting transposable elements and found that targeted TEs have higher levels of methylation relative to gene bodies and transcription start sites (TSS). Interestingly, these TEs are also enriched in CpGs compared to shuffled regions, even though the difference in the levels of methylation between CpG vs all regions of TEs does not look significantly different.

The study's description of transposable elements, piRNA pathway genes, piRNA populations, and piRNA source loci in axolotl is a valuable addition to the field, although the findings are not conceptually novel compared to studies in other animal model species. Since most of the study presents data on the axolotl piRNA pathway, the articles' framing on the evolution of a mammalian 'version' of the piRNA pathway lacks substantive support in the present form of the manuscript.

Major concerns

1. I find that there is a lack of congruence between the paper title + the first part of the abstract, which centres on mammalian piRNA pathway evolution, and the figure content, where five out of six main figures concern descriptions of the piRNA pathway and transposable elements in axolotl. The incongruence could be resolved by reframing the paper to be about axolotl or by expanding the evolutionary work.
2. The paper is framed on understanding the evolution of 'the mammalian version' of the piRNA pathway. However, apart from the presence of a PARN exonuclease paralog, the manuscript does not describe how such a mammalian version of the pathway would be functionally distinct from other versions of the pathway. Without a clear and functionally founded definition of a mammalian piRNA pathway, I believe the term adds confusion rather than clarity to the manuscript. The challenge of this term is also apparent in the abstract, where the authors write that "the mammalian version of the piRNA pathway [...] has been safeguarding the germline across most of tetrapod evolution." - i.e., well before the evolution of mammals. Reframing the evolutionary investigation to look at vertebrate piRNA pathway evolution could potentially resolve this issue.
3. The analyses of piRNA-pathway genes across multiple species seemingly relied exclusively on existing databases. Such annotations are generally not highly curated and provide a poor foundation for the analyses in Figure 1. Specifically: (a) unannotated piRNA pathway gene orthologs would be overlooked. (b) Because the evolutionary inferences hinge on only a

subset of nuclear-pathway components, it is crucial that these claims be supported with comprehensive homology validation. Both points could be addressed by performing searches (e.g., pBLAST and tBLASTn) for gene orthologs with additional reciprocal searches to establish 'reciprocal best hit' evidence to support orthology. Ideally, such analyses would be further substantiated by phylogenetic tree analyses. Furthermore, the list of genes that the authors look for is mainly rooted in the current knowledge of the mouse piRNA pathway. Looking beyond the mouse-like component list, e.g., searching for further Piwi/Dnmt/etc.-like genes, might uncover genes that functionally compensate for the seeming absence of some orthologs in certain species.

4. While authors have extended liberty to shape the direction of the Discussion section of a paper, the manuscript presents some very speculative models that do not help put the results in the right perspective. For example, at lines L373-L379, no reference is provided to support the impact of a possible energetic constraint on genome size. By contrast, recent papers do not support that metabolic rate relates to genome size (PMID: 31928192), but rather that for salamanders, the extent and speed of metamorphosis is a major parameter constraining genome size (PMID: 37143961). The proposal that efficient piRNA-mediated genome defence enabled the tetrapod water-to-land transition based on the inflated axolotl genome is very speculative and lacks support in the presented data and referenced literature. Similarly, the proposed connection between the mode of germ cell specification and the presence of nuclear piRNA-mediated transcriptional silencing would benefit from also considering species that have both germline preformation and nuclear PIWIs (e.g. roundworms and flies).

Minor concerns

5. Supplementary Figure 2A presents a complex Venn diagram with "The overlap of piRNA in the four samples". To understand the figure, it is important to clarify what the numbers and the overlap represent. E.g., do the piRNA numbers represent unique piRNA sequences or mapping positions?

6. Regarding the TE annotation (L190 and Figure 3C): how were the selected cutoffs for length and substitution percentage qualified? And what are the length cutoffs (missing from Methods)

7. Figure 4E-F: since many TE families are present in >1000 copies, how does the 1000 position multimapping are the shown piRNA tracks including multimapping piRNAs? Since 70%

8. Figure 4E: The small size of the panel tracks and the lack of track scales make it difficult to assess the data. This should be addressed by revision of the data presentation.

9. The characterization of piRNA clusters is somewhat shallow. For example: are clusters evolutionarily young TE insertions that happen to be close to each other in the genome, or are they rather structurally similar to either mouse or fly piRNA clusters? Here, it would also be helpful to have TE annotations in the 4F plots to show if the clusters overlap TEs or genes. Furthermore, connecting the axolotl piRNA cluster annotation to recent similar work from other non-model organisms would strengthen the work. One straightforward way to do this would be to apply the piCB tool (PMID: 39302833) and compare the results to those in that paper.

10. Line 212-213: When reporting the size profile of axolotl piRNAs, the recent similar description in Scharl et al 2024 (PMID: 39143221) should be cited.

Addition of non-essential suggestions

11. L68: This statement is missing a reference: "Among tetrapods, germline specification predominantly follows epigenesis, with the exception of frogs and birds, which utilize preformation."

12. Figure 6H and Supplementary Figure 5C: the box plots are very small, which makes them difficult to read

13. The authors show that while the active TEs in humans and mice have less DNAm than all TEs (Supplementary Figure 4), the inverse trend is observed in axolotl spermatozoa (Figure 6C). It would be good if the manuscript presented a possible explanation for this (technical or biological).

Referee #2:

General Assessment:

This manuscript presents a compelling and thorough evolutionary analysis of the piRNA pathway in vertebrates, identifying its modern mammalian form as having emerged earlier than previously understood. The authors combine phylogenetic analyses, transcriptomics, small RNA-seq, and methylome profiling—primarily in the axolotl model—to show that the cytoplasmic branch of the piRNA pathway started in lobe-finned fish and that both cytoplasmic and nuclear branches were fully present in early tetrapods.

The major strength of the manuscript is its depth of insights in evolutionary analysis. The phylogenetic mapping of cytoplasmic and nuclear piRNA pathway components across vertebrates is comprehensive and reveals important lineage-specific gains and losses. The authors' hypothesis linking nuclear piRNA pathway loss to preformation-based germline specification in birds and frogs is provocative and well-argued. In addition, the use of the axolotl—a model organism with a uniquely large genome and an ancestral mode of germline specification (epigenesis)—is particularly effective. The authors demonstrate piRNA pathway gene expression in both male and female germlines, robust piRNA biogenesis and amplification signatures, and evidence for piRNA-directed DNA methylation targeting young transposons. The multi-omic integration (small RNA-seq, EM-seq, transcriptomics, custom transposon annotation) and bioinformatic analyses are meticulous and of high quality.

This manuscript is a well-crafted, conceptually significant study that provides new insights into the evolution of epigenetic defenses in the germline. It represents a major contribution to our understanding of germline defense mechanisms and their role in shaping genome evolution. This reviewer recommends its publication in *The EMBO Journal* without major revision.

Before the manuscript goes to press, it would benefit from a revision addressing the following points:

Main comments:

1. The evidence for nuclear piRNA-mediated DNA methylation in axolotl is largely correlative (methylation over transposon promoters, CGI association), yet relevant claims in the manuscript often reach the level of functional interpretation. While direct functional assays (e.g., knockdown of *Pnldc1*, *PIWIL4*, *SPOCD1*) are understandably challenging in axolotl, the authors could strengthen their conclusions by explicitly discussing this limitation and refrain from over-interpreting the data.
2. Likewise, avoid overly functional claims for *Pnldc1* and other genes. For example, in the Abstract: "distinguished by usage of the 3' exonuclease *Pnldc1* for precursor piRNA trimming." The authors report that *Pnldc1* first appeared in the coelacanth genome but do not demonstrate whether it is expressed, functional, or utilized in the piRNA pathway in that species. The word "usage" overstates the evidence.
3. The manuscript may benefit from more clarification on germline specification correlation. The manuscript posits that the loss of the nuclear piRNA pathway correlates with germline preformation. While this hypothesis is attractive and plausible, the causality and evolutionary pressure behind this association remain speculative. A more cautious framing or discussion acknowledging alternative interpretations would strengthen the argument.
4. The authors note the lack of piRNA pathway components and canonical piRNAs in lungfish. Given the limited availability of lungfish genomic and transcriptomic datasets, further clarification on which datasets were used and how confident these inferences are would be helpful.

Minor Suggestions:

1. Abstract & Introduction: A brief mention of the relevance to genome size evolution would enhance reader engagement.
2. Maintain consistency in referring to species groups (e.g., "lobe-finned" vs. "lobe-fined").

Referee #3:

The manuscript, "On the origin of the mammalian piRNA pathway", is well written, the introduction to the piRNA pathway is concise, clear, and comprehensive.

But the report seems to have some confusion of identity. The title, abstract, and the figures suggest it is tackling the origins of the mammalian piRNA pathway. The main text suggests it is really about the piRNA pathway in the Axototl and the rest of the story hinges on this critical piece of data.

The study is being presented as a broad investigation into the origins of the piRNA pathway. This is fine as speculation, but a lot more evidence than a few "anecdotal" facts is needed to make the case. This part can be discussion, but not the main thrust of the paper? It is a matter of messaging, so I feel like it should be sold as an Axolotl paper with discussions centering on evolutionary implications etc.

I feel this is a major flaw that needs to be rectified. It could be two papers, or else say it is about the Axototl piRNA pathway.

overall, I like the paper and want it to be published, but with an altered focus.

A pet peeve, in these lines in the methods,

399-401 Small RNAs were excised from the gel and subjected to 3' linker ligation using pre-adenylated DNA linkers containing four random nucleotides at the 5' end to minimize ligation bias.

there is a need to cite a reference for this. It irks me sometimes that this contribution is not acknowledged in many publications (I am not worried about the review being anonymous!).

Jayaprakash AD, Jabado O, Brown BD, Sachidanandam R. Identification and remediation of biases in the activity of RNA ligases in small-RNA deep sequencing. *Nucleic Acids Res.* 2011 Nov;39(21):e141. doi: 10.1093/nar/gkr693. Epub 2011 Sep 2. PMID: 21890899; PMCID: PMC3241666.

Minor considerations: There is constant references to various organisms being "primitive" (already in coelocanth , or as early as axototl) in reference to the appearance of various components of the pathway (nuclear versus cytoplasmic machinery, exact molecular machinery etc.) This seems like a loose use of language that is a bit problematic. I think it is good to refer to a common ancestor etc .and talk about how the shared ancestor had it but was lost in some lineages etc. It might seem like a quibble, but I feel it is important to use language precisely when discussing evolution.

Referee #1:

We would like to thank this Reviewer for their support of our manuscript as well as their constructive and fair criticism. The points raised have been most insightful and extremely helpful in the revision of the manuscript. We hope the additional analysis, changes to the manuscript and the detailed response to the specific comments will satisfy the Reviewer's concerns. All changes to the manuscript are highlighted in blue.

Summary

Mammalian early development features a genome-wide demethylation phase during epigenetic reprogramming, which renders the germline genome susceptible to the activation and transposition of transposable elements (TEs). The piRNA pathway plays a key role in counteracting this threat by suppressing TEs and preserving genome integrity.

In this work, "On the origin of the mammalian piRNA pathway", the authors investigate the genetic framework and expression of piRNA genome defense pathway components in the salamander axolotl, identifying homologous piRNA pathway components conserved with the mouse system as a proxy for the "modern" mammalian pathway. They annotate the transposable element landscape in this species and analyze piRNA pools along with their genomic source loci, as well as stereotypical piRNA amplification 'ping-pong' signatures. From this, they identify piRNAs targeting transposable elements and found that targeted TEs have higher levels of methylation relative to gene bodies and transcription start sites (TSS). Interestingly, these TEs are also enriched in CpGs compared to shuffled regions, even though the difference in the levels of methylation between CpG vs all regions of TEs does not look significantly different.

The study's description of transposable elements, piRNA pathway genes, piRNA populations, and piRNA source loci in axolotl is a valuable addition to the field, although the findings are not conceptually novel compared to studies in other animal model species. Since most of the study presents data on the axolotl piRNA pathway, the articles' framing on the evolution of a mammalian 'version' of the piRNA pathway lacks substantive support in the present form of the manuscript.

Major concerns

1. I find that there is a lack of congruence between the paper title + the first part of the abstract, which centres on mammalian piRNA pathway evolution, and the figure content, where five out of six main figures concern descriptions of the piRNA pathway and transposable elements in axolotl. The incongruence could be resolved by reframing the paper to be about axolotl or by expanding the evolutionary work.

We agree that the title and parts of the abstract did not accurately reflect the focus of the paper. To reflect this, we have changed the title to '*A mammalian-like piRNA pathway in Axolotl reveals the origins of piRNA-directed DNA methylation*'. We hope this title accurately captures the axolotl focus and the key evolutionary implications. Furthermore, the abstract is more focused and substantially shortened to comply with the 150 words *EMBO Journal* format. We have completely rewritten the abstract with a focus on the nuclear piRNA pathway. We have emphasized that piRNA-directed DNA methylation, as distinct from the cytoplasmic pathway, is believed to be bespoke to mammals. This emphasis and clarity now really highlight the core findings of the manuscript and their importance. We hope these changes to the title and abstract address this important point to bring an axolotl focus to the manuscript. The extended evolutionary implications are left to the discussion. In summary, we hope these combined changes address this concern.

2. The paper is framed on understanding the evolution of 'the mammalian version' of the piRNA pathway. However, apart from the presence of a PARN exonuclease paralog, the manuscript does not describe how such a mammalian version of the pathway would be functionally distinct from other versions of the pathway. Without a clear and functionally founded definition of a mammalian piRNA pathway, I believe the term adds confusion rather than clarity to the manuscript. The challenge of this term is also apparent in the abstract, where the authors write that "the mammalian version of the piRNA pathway [...] has been safeguarding the germline across most of tetrapod evolution." - i.e., well before the evolution of mammals. Reframing the evolutionary investigation to look at vertebrate piRNA pathway evolution could potentially resolve this issue.

We thank the reviewer for this thoughtful and important comment. We have made some changes to the introduction, placing the following definition of the mammalian nuclear piRNA pathway: '*In summary, the factors PIWIL4, SPOCD1, TEX15, DNMT3B/C and C19ORF84 that mediate piRNA-directed DNA methylation define the mammalian nuclear piRNA pathway.*'

We have completely rewritten the abstract with a focus on the nuclear piRNA pathway. The abstract is more focused and substantially shortened to comply with the 150 words *EMBO Journal* format. We have emphasized that piRNA-directed DNA methylation, as distinct from the cytoplasmic pathway, is believed to be bespoke to mammals.

We hope these changes to the title, abstract and introduction address this important point to clarify the core novelty and advancement of the manuscript.

3. The analyses of piRNA- pathway genes across multiple species seemingly relied exclusively on existing databases. Such annotations are generally not highly curated and provide a poor foundation for the analyses in Figure 1. Specifically: (a) unannotated piRNA pathway gene orthologs would be overlooked. (b) Because the evolutionary inferences hinge on only a subset of nuclear- pathway components, it is crucial that these claims be supported with comprehensive homology validation. Both points could be addressed by performing searches (e.g., pBLAST and tBLASTn) for gene orthologs with additional reciprocal searches to establish 'reciprocal best hit' evidence to support orthology. Ideally, such analyses would be further substantiated by phylogenetic tree analyses. Furthermore, the list of genes that the authors look for is mainly rooted in the current knowledge of the mouse piRNA pathway. Looking beyond the mouse-like component list, e.g., searching for further Piwi/Dnmt/etc.-like genes, might uncover genes that functionally compensate for the seeming absence of some orthologs in certain species.

We thank the reviewer for this great comment. We had performed 'reciprocal best hit' analysis using tBLASTn searches to identify potential unannotated genes that were initially missing from genome annotations. However, these additional searches did not yield any new gene identifications. As this analysis produced only negative results, we did not include it in the Methods section. Now the Methods have been updated accordingly to include the BLAST analysis. The genomes included in our analysis are from species with high-quality assemblies and annotations. We initially considered incorporating additional intermediate species between fish and tetrapods, such as bichir, sturgeons, bowfin, and caecilians. However, the quality of genome assemblies and annotations for these species was insufficient to reliably identify piRNA pathway genes. To ensure the robustness of our conclusions, we therefore keep our analysis to species with well-annotated genomes, shown in Fig. 1.

We have included all known homologs for the *Dnmt* and mammalian *Piwi* genes (*Dnmt3a*, *Dnmt3b*, *Dnmt3c*, *Dnmt3l*, *Piwil1*, *Piwil2*, *Piwil3* and *Piwil4*) in our analyses. The cytoplasmic PIWIL2 gene is orthologues of *Drosophila* Ago3, while mammalian PIWIL1/PIWIL3/PIWIL4

are not related to *Drosophila* Piwi, they arose independently^{1,2}. *Drosophila* does not utilize DNA methylation to silence transposons but rather heterochromatinization through histone modification³⁻⁸. Panx is the key Piwi-associated factor that mediates piRNA-PIWI-mediated transposon co-transcription silencing and downstream heterochromatinization^{6,7}. Panx is not conserved in mammals⁷. Panx forms a complex with the nuclear export factors Nxt1 and Nxf1 as well as the dynein Ctp^{3-5,8-10}. While orthologues of these proteins exist in mammals^{4,9}, none of them have been found to be associated with PIWIL4, SPOCD1, C19ORF84, DNMT3L and DNMT3C^{11,12}. In summary, *Drosophila* Piwi and Panx are not found in mammals, they mediate a different silencing process, and their associated factors (Nxt1, Nxf1 and Ctp) are not found to be associated with mammalian nuclear piRNA factors¹⁻¹⁰. We are therefore confident of the approach that we have taken will identify key vertebrate nuclear piRNA factors.

We hope the additions to the manuscript and the above clarifications address these important concerns.

4. While authors have extended liberty to shape the direction of the Discussion section of a paper, the manuscript presents some very speculative models that do not help put the results in the right perspective. For example, at lines L373-L379, no reference is provided to support the impact of a possible energetic constraint on genome size. By contrast, recent papers do not support that metabolic rate relates to genome size (PMID: 31928192), but rather that for salamanders, the extent and speed of metamorphosis is a major parameter constraining genome size (PMID: 37143961). The proposal that efficient piRNA-mediated genome defence enabled the tetrapod water-to-land transition based on the inflated axolotl genome is very speculative and lacks support in the presented data and referenced literature.

We are extremely grateful for highlighting recent literature that runs counter to our speculation on metabolic constraints associated with genome size. Our correlation was naively intuitive to us. In the light of these publications, we have removed our speculation in the discussion. Specifically, we have removed the following sentences in the discussion: '*Due to its ecological niche and metabolic requirements, the axolotl can tolerate the extra energetic demands and larger cell sizes associated with a giant genome, and as such may not experience strong selective pressure to reduce its genome size.*' and '*The transition to land probably placed greater energetic constraints on terrestrial tetrapods and hence a pressure to reduce genome sizes.*'. Again, we are extremely grateful for this insight.

Similarly, the proposed connection between the mode of germ cell specification and the presence of nuclear piRNA-mediated transcriptional silencing would benefit from also considering species that have both germline preformation and nuclear PIWIs (e.g. roundworms and flies).

The correlation we made between germline specification and a functional nuclear piRNA pathway was limited to tetrapods. To be very clear, we have now emphasized this in the discussion. We have also highlighted that preformation can associate with a functional nuclear piRNA pathway in invertebrates.

Minor concerns

5. Supplementary Figure 2A presents a complex Venn diagram with "The overlap of piRNA in the four samples". To understand the figure, it is important to clarify what the numbers and the overlap represent. E.g., do the piRNA numbers represent unique piRNA sequences or mapping positions?

The piRNA numbers represent unique piRNA sequences. We have changed the figure legend for clarification.

6. Regarding the TE annotation (L190 and Figure 3C): how were the selected cutoffs for length and substitution percentage qualified? And what are the length cutoffs (missing from Methods)

We acknowledge that both length and divergence thresholds are approximations and may vary among transposon families. Nevertheless, we have applied stringent criteria to ensure robustness of our classification. The length cutoffs were determined based on previous publications about each transposon family. The publications used to estimate the length cutoff for each transposon family as well as the cutoff length are now detailed in Methods and Table EV2). Substitution percentage (percDiv) was set based on our lab's prior work¹¹ as well as other studies identifying active transposons¹³. Young transposons with <5% percDiv (equivalent to <50 milliDiv) exhibit dramatic loss of DNA methylation upon knockout of piRNA pathway factors¹¹, supporting this threshold as a marker of transposon activity.

7. Figure 4E-F: since many TE families are present in >1000 copies, how does the 1000 position multimapping are the shown piRNA tracks including multimapping piRNAs?

The majority of piRNAs could be mapped up to 500 genomic sites (Reviewer Fig. 1). Therefore, allowing up to 1000 multi-mapping positions would ensure accurate alignment of piRNAs. For tracks in Fig. 4e-f, 1/n weighting strategy was applied to distribute the contribution of each piRNA mapped sites. For example, if a piRNA sequence mapped to 100 genomic sites, each site was assigned a score of 0.01. The signal for each genomic bin was calculated as the sum of all piRNA scores within that bin, followed by RPM normalization and log2 transformation for visualization. Details of this approach were added in the Methods section.

Figure for reviewers removed

To further address this concern and ensure the robustness of our results, we compared piRNA profiles generated using multi-mapping limits of 100, 1000, and 5000. The results were consistent across thresholds (Reviewer Fig. 2), indicating that the observed patterns are not sensitive to the specific multi-mapping setting.

Figure for reviewers removed

8. Figure 4E: The small size of the panel tracks and the lack of track scales make it difficult to assess the data. This should be addressed by revision of the data presentation.

Thank you for this suggestion. Fig. 4e has been re-plotted to address this point. Track scales have been added. Fig. 4e has been moved and now is found in Fig. 5a.

9. The characterization of piRNA clusters is somewhat shallow. For example: are clusters evolutionarily young TE insertions that happen to be close to each other in the genome, or are they rather structurally similar to either mouse or fly piRNA clusters?

The axolotl piRNA clusters are structurally like the mouse. In summary, the axolotl piRNA clusters show the following three key features: the majority are uni-strand (Fig. 4e); they have a median length of 10 kb (Fig. 4g); and piRNAs are predominantly derived from a small number of highly productive clusters (Fig. 4f). This is consistent with the unidirectional transcription mechanism typical of mammalian piRNA clusters like mouse¹⁴⁻¹⁷. In contrast, most piRNAs in *Drosophila* are from dual-stranded transcribed piRNA clusters¹⁸. The length distribution of piRNA clusters in the axolotl is also similar to mouse¹⁴⁻¹⁷. Similarly, as in other vertebrate and invertebrate species¹⁹, piRNAs are mainly derived from a few major clusters.

Here, it would also be helpful to have TE annotations in the 4F plots to show if the clusters overlap TEs or genes. Furthermore, connecting the axolotl piRNA cluster annotation to recent similar work from other non-model organisms would strengthen the work.

Transposon tracks from both forward and reverse stand have been added in Fig. 4f (new Fig. 5b). The piRNA clusters overlapped with transposons, especially for LTR, DNA and LINE transposons.

One straightforward way to do this would be to apply the piCB tool (PMID: 39302833) and compare the results to those in that paper.

We also thank the reviewer for recommending the piCB paper, which we found to be highly insightful. We attempted to apply the piCB tool as suggested; however, due to the exceptionally large genome size of axolotl, it was not feasible to process the data using piCB. piCB imposes a maximum chromosome length of 536 Mbp, while most axolotl chromosomes exceed this limit, with the longest reaching 1.63 Gb. As an alternative approach, we referred to the key findings of the piCB study and performed comparable analyses using the proTRAC tool (Table EV4, Fig. 4e-h), which is already employed in our manuscript. Further additional piRNA cluster analyses are now found in the main text.

10. Line 212-213: When reporting the size profile of axolotl piRNAs, the recent similar description in Scharl et al 2024 (PMID: 39143221) should be cited.

We thank the reviewer for this suggestion. The description in Scharl et al. 2024 has been cited.

Addition of non-essential suggestions

11. L68: This statement is missing a reference: "Among tetrapods, germline specification predominantly follows epigenesis, with the exception of frogs and birds, which utilize preformation."

The citation has been added appropriately.

12. Figure 6H and Supplementary Figure 5C: the box plots are very small, which makes them difficult to read.

We thank the reviewer for this suggestion. The figures have been re-plotted with clear display (new Fig. 7h, Fig. EV4d).

13. The authors show that while the active TEs in humans and mice have less DNAm than all TEs (Supplementary Figure 4), the inverse trend is observed in axolotl spermatozoa (Figure 6C). It would be good if the manuscript presented a possible explanation for this (technical or biological).

We thank the reviewer for pointing this out. We have investigated this in great detail. Indeed, active transposons in mouse and human have less methylation across the entire element compared to all transposons. However, promoter regions are the target of the piRNA-directed DNA methylation in mouse^{11,12,20}. We observe higher promoter methylation in the mouse but not humans (Reviewer Fig. 3). We are unsure of the reason for the slightly lower methylation of active LINE1s in human spermatozoa. It could be technical or real, and if real likely inconsequential to transposon silencing. Should the difference be real, it is likely inconsequential as we have shown that *SPOCD1* deficiency in humans results in male infertility and LINE1 expression¹². As we are unsure of the reason and the relevance of slightly lower DNA methylation of active human LINE1 promoters, we propose to remove mouse and human comparison from the manuscript.

Figure for reviewers removed

Referee #2:

We would like to thank this Reviewer for their support of our manuscript as well as their constructive and fair criticism. The points raised have been most insightful and extremely helpful in the revision of the manuscript. We hope the additional analysis, changes to the manuscript and the detailed response to the specific comments will satisfy the Reviewer's concerns. All changes to the manuscript are highlighted in blue.

General Assessment:

This manuscript presents a compelling and thorough evolutionary analysis of the piRNA pathway in vertebrates, identifying its modern mammalian form as having emerged earlier than previously understood. The authors combine phylogenetic analyses, transcriptomics, small RNA-seq, and methylome profiling—primarily in the axolotl model—to show that the cytoplasmic branch of the piRNA pathway started in lobe-finned fish and that both cytoplasmic and nuclear branches were fully present in early tetrapods.

The major strength of the manuscript is its depth of insights in evolutionary analysis. The phylogenetic mapping of cytoplasmic and nuclear piRNA pathway components across vertebrates is comprehensive and reveals important lineage-specific gains and losses. The authors' hypothesis linking nuclear piRNA pathway loss to preformation-based germline specification in birds and frogs is provocative and well-argued. In addition, the use of the axolotl—a model organism with a uniquely large genome and an ancestral mode of germline specification (epigenesis)—is particularly effective. The authors demonstrate piRNA pathway gene expression in both male and female germlines, robust piRNA biogenesis and amplification signatures, and evidence for piRNA-directed DNA methylation targeting young transposons. The multi-omic integration (small RNA-seq, EM-seq, transcriptomics, custom transposon annotation) and bioinformatic analyses are meticulous and of high quality.

This manuscript is a well-crafted, conceptually significant study that provides new insights into the evolution of epigenetic defenses in the germline. It represents a major contribution to our understanding of germline defense mechanisms and their role in shaping genome evolution. This reviewer recommends its publication in *The EMBO Journal* without major revision.

Before the manuscript goes to press, it would benefit from a revision addressing the following points:

Main comments:

1. The evidence for nuclear piRNA-mediated DNA methylation in axolotl is largely correlative (methylation over transposon promoters, CGI association), yet relevant claims in the manuscript often reach the level of functional interpretation. While direct functional assays (e.g., knockdown of *Pnldc1*, *PIWIL4*, *SPOCD1*) are understandably challenging in axolotl, the authors could strengthen their conclusions by explicitly discussing this limitation and refrain from over-interpreting the data.

We thank the reviewer for this thoughtful comment. To address this point, we have added the following sentence in the discussion: *'However, genetic studies will be required to formally demonstrate a functional role for piRNA-directed transposon methylation in the axolotl germline.'*

2. Likewise, avoid overly functional claims for *Pnldc1* and other genes. For example, in the Abstract: "distinguished by usage of the 3' exonuclease *Pnldc1* for precursor piRNA trimming." The authors report that *Pnldc1* first appeared in the coelacanth genome but do

not demonstrate whether it is expressed, functional, or utilized in the piRNA pathway in that species. The word "usage" overstates the evidence.

We have revised the manuscript to moderate our claims and not infer usage but rather state presence. Regarding *Pnlc1*, we have completely rewritten the abstract with a focus on the nuclear piRNA pathway to highlight the core novelty and advancement of the manuscript. We have also changed the word "usage" to "presence" in the discussion. In summary the term "usage" is no longer present in the manuscript. We are very grateful for this comment.

3. The manuscript may benefit from more clarification on germline specification correlation. The manuscript posits that the loss of the nuclear piRNA pathway correlates with germline preformation. While this hypothesis is attractive and plausible, the causality and evolutionary pressure behind this association remain speculative. A more cautious framing or discussion acknowledging alternative interpretations would strengthen the argument.

We thank the reviewer for this great suggestion. To address this point, we have added the following sentence in the discussion: '*While we do not understand the basis of the correlation between the presence of a nuclear piRNA pathway and the usage of epigenesis for germline specification in tetrapods, it might be linked to the evolution epigenetic germline reprogramming.*'. We have also highlighted that several invertebrate species that use preformation have a nuclear piRNA pathway. In summary, we have expanded the discussion on this aspect of the manuscript. We hope these additions to the manuscript address this important point.

4. The authors note the lack of piRNA pathway components and canonical piRNAs in lungfish. Given the limited availability of lungfish genomic and transcriptomic datasets, further clarification on which datasets were used and how confident these inferences are would be helpful.

The cited papers^{21,22} present the most comprehensive and up-to-date annotation for three lungfish genomes. While we acknowledge the potential limitation on the current lungfish genome annotation, we are confident of the overall finding that lungfish have lost many piRNA pathway genes. Our confidence stems from the seminal findings that the South American and African lungfish have lost canonically sized piRNAs that correlates with their enormous and expanding genomes²¹.

Minor Suggestions:

1. Abstract & Introduction: A brief mention of the relevance to genome size evolution would enhance reader engagement.

We thank the reviewer for this suggestion. We have added the relevance to genome size evolution in introduction.

2. Maintain consistency in referring to species groups (e.g., "lobe-finned" vs. "lobe-fined").

We thank the reviewer for this suggestion. We have made the term consistent throughout the manuscript.

Referee #3:

We would like to thank this Reviewer for their support of our manuscript as well as their constructive and fair criticism. The points raised have been most insightful and extremely helpful in the revision of the manuscript. We hope the additional analysis, changes to the manuscript and the detailed response to the specific comments will satisfy the Reviewer's concerns. All changes to the manuscript are highlighted in blue.

The manuscript, "On the origin of the mammalian piRNA pathway", is well written, the introduction to the piRNA pathway is concise, clear, and comprehensive.

But the report seems to have some confusion of identity. The title, abstract, and the figures suggest it is tackling the origins of the mammalian piRNA pathway. The main text suggests it is really about the piRNA pathway in the Axototl and the rest of the story hinges on this critical piece of data.

The study is being presented as a broad investigation into the origins of the piRNA pathway. This is fine as speculation, but a lot more evidence that a few "anecdotal" facts is needed to make the case. This part can be discussion, but not the main thrust of the paper ? It is a matter of messaging, so I feel like it should be sold as a Axolotl paper with discussions centering on evolutionary implications etc. I feel this is a major flaw that needs to be rectified. It could be two papers, or else say it is about the Axototl piRNA pathway.

We agree that the trust of the paper was not aligned with the fact that the bulk of data being from investigations of the axolotl genome and germline. To reflect this, we have changed the title to '*A mammalian-like piRNA pathway in Axolotl reveals the origins of piRNA-directed DNA methylation*'. We hope this title accurately captures the axolotl focus and the key evolutionary implication. Furthermore, the abstract is more focused and substantially shortened to comply with the 150 words *EMBO Journal* format. We have completely rewritten the abstract with a focus on the nuclear piRNA pathway. We have emphasized that piRNA-directed DNA methylation, as distinct from the cytoplasmic pathway, is believed to be bespoke to mammals. This emphasis and clarity now really highlight the core findings of the manuscript and their importance. We hope these changes to the title, abstract and manuscript address this important point to bring a more axolotl focus to the manuscript. The extended evolutionary implications are left to the discussion. In summary, we hope these combined changes address your central concern.

overall, I like the paper and want it to be published, but with an altered focus.

A pet peeve, in these lines in the methods,

399-401 Small RNAs were excised from the gel and subjected to 3' linker ligation using pre-adenylated DNA linkers containing four random nucleotides at the 5' end to minimize ligation bias. there is a need to cite a reference for this. It irks me sometimes that this contribution is not acknowledged in many publications (I am not worried about the review being anonymous !).

Jayaprakash AD, Jabado O, Brown BD, Sachidanandam R. Identification and remediation of biases in the activity of RNA ligases in small-RNA deep sequencing. *Nucleic Acids Res.* 2011 Nov;39(21):e141. doi: 10.1093/nar/gkr693. Epub 2011 Sep 2. PMID: 21890899; PMCID: PMC3241666.

We apologize for missing this important citation. We have now cited this work appropriately in the revised manuscript. We are extremely grateful for bringing this omission to our attention.

Minor considerations: There is constant references to various organisms being "primitive" (already in coelocanth , or as early as axototl) in reference to the appearance of various components of the pathway (nuclear versus cytoplasmic machinery, exact molecular machinery etc.) This seems like a loose use of language that is a bit problematic. I think it is good to refer to a common ancestor etc .and talk about how the shared ancestor had it but was lost in some lineages etc. It might seem like a quibble, but I feel it is important to use language precisely when discussing evolution.

We thank the reviewer for this great suggestion. We have done our best to refine the evolutionary language throughout the text.

References

1. Jehn, J. *et al.* PIWI genes and piRNAs are ubiquitously expressed in mollusks and show patterns of lineage-specific adaptation. *Commun Biol* **1**, 1–11 (2018).
2. Kerner, P., Degnan, S. M., Marchand, L., Degnan, B. M. & Vervoort, M. Evolution of RNA-Binding Proteins in Animals: Insights from Genome-Wide Analysis in the Sponge *Amphimedon queenslandica*. *Molecular Biology and Evolution* **28**, 2289–2303 (2011).
3. Batki, J. *et al.* The nascent RNA binding complex SFiNX licenses piRNA-guided heterochromatin formation. *Nat Struct Mol Biol* **26**, 720–731 (2019).
4. Fabry, M. H. *et al.* piRNA-guided co-transcriptional silencing coopts nuclear export factors. *eLife* **8**, e47999 (2019).
5. Murano, K. *et al.* Nuclear RNA export factor variant initiates piRNA- guided co-transcriptional silencing. *The EMBO Journal* **38**, e102870 (2019).
6. Sienski, G. *et al.* Silencio/CG9754 connects the Piwi-piRNA complex to the cellular heterochromatin machinery. *Genes Dev* **29**, 2258–2271 (2015).
7. Yu, Y. *et al.* Panoramix enforces piRNA-dependent cotranscriptional silencing. *Science* **350**, 339–342 (2015).
8. Zhao, K. *et al.* A Pandas complex adapted for piRNA-guided transcriptional silencing and heterochromatin formation. *Nat Cell Biol* **21**, 1261–1272 (2019).
9. Eastwood, E. L. *et al.* Dimerisation of the PICTS complex via LC8/Cut-up drives co-transcriptional transposon silencing in *Drosophila*. *eLife* **10**, e65557 (2021).
10. Schnabl, J. *et al.* Molecular principles of Piwi-mediated cotranscriptional silencing through the dimeric SFiNX complex. *Genes Dev.* **35**, 392–409 (2021).
11. Zoch, A. *et al.* SPOCD1 is an essential executor of piRNA-directed de novo DNA methylation. *Nature* **584**, 635–639 (2020).
12. Zoch, A. *et al.* C19ORF84 connects piRNA and DNA methylation machineries to defend the mammalian germ line. *Molecular Cell* **84**, 1021-1035.e11 (2024).

13. Novick, P. A., Smith, J. D., Floumanhaft, M., Ray, D. A. & Boissinot, S. The Evolution and Diversity of DNA Transposons in the Genome of the Lizard *Anolis carolinensis*. *Genome Biology and Evolution* **3**, 1–14 (2011).
14. Girard, A., Sachidanandam, R., Hannon, G. J. & Carmell, M. A. A germline-specific class of small RNAs binds mammalian Piwi proteins. *Nature* **442**, 199–202 (2006).
15. Aravin, A. *et al.* A novel class of small RNAs bind to MILI protein in mouse testes. *Nature* **442**, 203–207 (2006).
16. Grivna, S. T., Beyret, E., Wang, Z. & Lin, H. A novel class of small RNAs in mouse spermatogenic cells. *Genes Dev* **20**, 1709–1714 (2006).
17. Li, X. Z. *et al.* An Ancient Transcription Factor Initiates the Burst of piRNA Production during Early Meiosis in Mouse Testes. *Molecular Cell* **50**, 67–81 (2013).
18. Brennecke, J. *et al.* Discrete Small RNA-Generating Loci as Master Regulators of Transposon Activity in *Drosophila*. *Cell* **128**, 1089–1103 (2007).
19. Konstantinidou, P. *et al.* A comparative roadmap of PIWI-interacting RNAs across seven species reveals insights into *de novo* piRNA-precursor formation in mammals. *Cell Reports* **43**, 114777 (2024).
20. Dias Mirandela, M. *et al.* Two-factor authentication underpins the precision of the piRNA pathway. *Nature* **634**, 979–985 (2024).
21. Schartl, M. *et al.* The genomes of all lungfish inform on genome expansion and tetrapod evolution. *Nature* **634**, 96–103 (2024).
22. Wang, K. *et al.* African lungfish genome sheds light on the vertebrate water-to-land transition. *Cell* **184**, 1362-1376.e18 (2021).

Dear Dr. O'Carroll,

Thank you for submitting your revised manuscript for consideration by the EMBO Journal.

I would like to start by apologizing for the long time it took us to send you this decision letter. We have spent the last month chasing Referee #2 with no success and eventually decided to continue without their feedback.

Your manuscript has now been seen by two of the three original referees whose comments are enclosed. As you will see, both referees express interest in your manuscript and are broadly in favour of publication, pending satisfactory minor revision. Moreover, I include below the specific comments by our editorial assistance team that go over the more technical aspects of manuscripts. It is very important you will also take their comments into your revision plan as these are essential for final acceptance and publication.

Given the referees' positive recommendations, I would like to invite you to submit a revised version of the manuscript, addressing the comments of both reviewers.

There is no need to address in the response letter the editorial comments by our assistance team (yet, please make sure to revise your manuscript to solve the issues they raise).

We generally allow three months as standard revision time. Yet, I am confident you will not need such a long time for submitting your revised manuscript in light of the relatively minor comments.

Thank you for the opportunity to consider your work for publication. I look forward to your revision.

Yours sincerely,

Yehu Moran
Academic Editor
The EMBO Journal

Please remember: Digital image enhancement is acceptable practice, as long as it accurately represents the original data and conforms to community standards. If a figure has been subjected to significant electronic manipulation, this must be noted in the

figure legend or in the 'Materials and Methods' section. The editors reserve the right to request original versions of figures and the original images that were used to assemble the figure.

We realize that it is difficult to revise to a specific deadline. In the interest of protecting the conceptual advance provided by the work, we recommend a revision within 3 months (30th Dec 2025). Please discuss the revision progress ahead of this time with the editor if you require more time to complete the revisions.

Specific comments by editorial assistance team

*DATA AVAILABILITY SECTION: Please add the Code Availability under Data Availability

*Author Contributions: Please remove from manuscript text and keep only in our submission system.

*DisclCIS: rename "Disclosure and competing interests statement"

*REFERENCE FORMAT: In the text of the manuscript, a reference should be cited by author and year of publication; no more than two authors may be cited per reference; 'et al' should be used if there are more than two authors (i.e. Smith & Jones, 2003; Smith et al, 2000). In the reference list, citations should be listed in alphabetical order and then chronologically, with the authors' surnames and initials inverted; where there are more than 10 authors on a paper, 10 will be listed, followed by 'et al.'.

*DATASET EV LEGENDS: Please rename Tables EV1 and EV2 and make them Dataset EV1 and EV2. Please add a legend to each dataset containing the title and a short description, and place it in a separate tab/worksheet. Please update the numbering of the remaining EV tables accordingly, and add a legend to the top of each table.

*SOURCE DATA: Please provide the numerical data for Fig 3C, Fig 5A,B as an .xlsx or .csv file.

*SYNOPSIS IMAGE: Please provide (you can look on recently published papers as an example).

*SYNOPSIS TEXT: Please provide (you can look on recently published papers as an example).

Extra Notes: Please update the section heading to "Methods"

- Figure legends:

1. Please note that the exact p values are not provided in the legends of figures 7C, G. Please correct.

2. Please note that information related to n is missing in the legends of figures 7A, C, G, H; EV4 A, D. Please correct.

Referee comments

Referee #1:

The authors have addressed all my concerns sufficiently (see details below) and the manuscript stands much improved.

Regarding major concerns #1 and #2

The revised title and abstract now align well with the paper's findings. Together with the revised description of a mammalian-type piRNA pathway in the introduction, these changes effectively address the points I raised under major concerns #1 and #2 about the paper angle being misaligned with the presented data.

Minor comment: the new sentence "a mammalian-like piRNA pathway [...] is already found and expressed in its current configuration in the axolotl salamander" is imprecise, as 'already' - to me - should refer to the last common ancestor of axolotl and mice.

Regarding major concern #3

This concern is well addressed by the added clarification of the analyses performed to ensure rigorous and comprehensive detection of piRNA pathway genes.

Regarding major concern #4

The revised discussion is much improved.

Regarding the minor concerns

All minor concerns have been well addressed in the revised manuscript and through the clarification in the point-by-point response, including the careful further analyses provided in the reviewer figures 1-3.

Referee #3:

I think the authors have addressed all the concerns of the reviewers from the previous round old reviews. I would suggest the title be toned down a little bit ? "A mammalian-like piRNA pathway in Axolotl [reveals] the origins of piRNA-directed DNA methylation" instead of reveals, maybe something a bit less definitive ? such as "indicates" perhaps ? I think this carried over from the previous version.
otherwise no other suggestions or criticisms.

Referee #1:

The authors have addressed all my concerns sufficiently (see details below) and the manuscript stands much improved.

Regarding major concerns #1 and #2

The revised title and abstract now align well with the paper's findings. Together with the revised description of a mammalian-type piRNA pathway in the introduction, these changes effectively address the points I raised under major concerns #1 and #2 about the paper angle being misaligned with the presented data.

Minor comment: the new sentence "a mammalian-like piRNA pathway [...] is already found and expressed in its current configuration in the axolotl salamander" is imprecise, as 'already' - to me - should refer to the last common ancestor of axolotl and mice.

We thank the reviewer for the thoughtful comment. We have now removed the word "already" to avoid any imprecision.

Regarding major concern #3

This concern is well addressed by the added clarification of the analyses performed to ensure rigorous and comprehensive detection of piRNA pathway genes.

Regarding major concern #4

The revised discussion is much improved.

Regarding the minor concerns

All minor concerns have been well addressed in the revised manuscript and through the clarification in the point-by-point response, including the careful further analyses provided in the reviewer figures 1-3.

Referee #3:

I think the authors have addressed all the concerns of the reviewers from the previous round old reviews. I would suggest the title be toned down a little bit? "A mammalian-like piRNA pathway in Axolotl [reveals] the origins of piRNA-directed DNA methylation" instead of reveals, maybe something a bit less definitive? such as "indicates" perhaps? I think this carried over from the previous version. I otherwise no other suggestions or criticisms.

We agree. We have revised the title: "A mammalian-like piRNA pathway in Axolotl indicates the origins of piRNA-directed DNA methylation."

Dear Dr. O'Carroll,

I am pleased to inform you that your manuscript has been accepted for publication in the EMBO Journal.

Yours sincerely,

Yehu Moran
Academic Editor
The EMBO Journal
